# Sustained CREB phosphorylation by lipid-peptide liquid crystalline nanoassemblies

Yu Wu[1], Borislav Angelov [2✉], Yuru Deng[3], Takehiko Fujino[4], Md Shamim Hossain[4], Markus Drechsler [5] & Angelina Angelova [1✉]

Cyclic-AMP-response element-binding protein (CREB) is a leucine zipper class transcription factor that is activated through phosphorylation. Ample CREB phosphorylation is required for neurotrophin expression, which is of key importance for preventing and regenerating neurological disorders, including the sequelae of long COVID syndrome. Here we created lipid-peptide nanoassemblies with different liquid crystalline structural organizations (cubosomes, hexosomes, and vesicles) as innovative nanomedicine delivery systems of bioactive PUFA-plasmalogens (vinyl ether phospholipids with polyunsaturated fatty acid chains) and a neurotrophic pituitary adenylate cyclase-activating polypeptide (PACAP). Considering that plasmalogen deficiency is a potentially causative factor for neurodegeneration, we examined the impact of nanoassemblies type and incubation time in an in vitro Parkinson's disease (PD) model as critical parameters for the induction of CREB phosphorylation. The determined kinetic changes in CREB, AKT, and ERK-protein phosphorylation reveal that non-lamellar PUFA-plasmalogen-loaded liquid crystalline lipid nanoparticles significantly prolong CREB activation in the neurodegeneration model, an effect unattainable with free drugs, and this effect can be further enhanced by the cell-penetrating peptide PACAP. Understanding the sustained CREB activation response to neurotrophic nanoassemblies might lead to more efficient use of nanomedicines in neuroregeneration.

[1] Université Paris-Saclay, CNRS, Institut Galien Paris-Saclay, F-91400 Orsay, France. [2] Extreme Light Infrastructure ERIC, CZ-25241 Dolni Brezany, Czech Republic. [3] Wenzhou Institute, University of Chinese Academy of Sciences, No.1, Jinlian Road, Longwan District, Wenzhou, Zhejiang 325001, China. [4] Institute of Rheological Functions of Food, 2241-1 Kubara, Hisayama-cho, Kasuya-gun, Fukuoka 811-2501, Japan. [5] Keylab "Electron and Optical Microscopy", Bavarian Polymerinstitute (BPI), University of Bayreuth, Universitätsstrasse 30, D-95440 Bayreuth, Germany. ✉email: Borislav.Angelov@eli-beams.eu; angelina.angelova@universite-paris-saclay.fr

Lipid nanoparticles (LNPs) are one of the most successful nanocarrier types in nanomedicine development[1–5]. They comprise safe, controlled drug delivery systems for therapeutic intervention in various pathologies (e.g., bacterial and viral infections, cancer, neurological, psychiatric, skin, ocular and other diseases)[6–12]. The efficacy of LNP-based mRNA vaccines against the coronavirus disease COVID-19 has exceeded 95%[13]. An emerging direction in nanomedicine research focuses on the strategies for recovery from neurological disorders linked to long COVID-19 syndrome[14,15]. Since 2020, accumulating evidence has revealed that SARS-CoV-2 coronavirus invades the central neural system (CNS) and may provoke chronic neuroinflammation and neurodegeneration in vulnerable post-COVID-19 patients[16–19]. As a result, the neurological sequelae of long COVID-19 syndrome tend to significantly augment the prevalence of neurodegenerative disorders (NDs)[20–22], including Alzheimer's disease (AD), Parkinson's disease (PD), amyotrophic lateral sclerosis and Huntington's disease, for which there is no efficient treatment yet.

Although the understanding of long COVID-19 pathology is still limited, the role of the cyclic AMP (cAMP)-response element-binding protein (CREB) can be highlighted as crucial for the regeneration of CNS or peripheral neuronal damages[23,24]. Neurodegeneration and neuronal cell death may be due to multiple factors, e.g., neurotrophin deficiency, altered lipid metabolism, oxidative stress, mitochondrial dysfunction, DNA damage, protein misfolding and aggregation, etc[25–27]. The key function of CREB has been emphasized in the search of new strategies against neurodegeneration at cellular and subcellular levels[28]. This transcription factor regulates the expression of genes and neurotrophic proteins e.g., brain-derived neurotrophic factor, which are of particular importance for nanomedicine-stimulated adult neurogenesis, neuronal plasticity, and regeneration[29]. Of note, CREB also regulates the cellular response to oxidative stress[30]. Experimental evidence has confirmed that CREB is activated through phosphorylation at Ser-133, the main site for regulating its transcriptional activity[31]. In general, CREB can be phosphorylated by different stimuli, e.g., by cAMP, calcium, stress-activated MAPK, or growth factors. The activation (phosphorylation) of CREB is required for neuronal cell survival and proliferation, and therefore, for improvement of cognitive function[32]. However, clinical studies have established essentially reduced CREB phosphorylation in AD patients[33]. Therapeutic developments have pointed out that certain natural substances, which exert neuroprotective effects (e.g., curcumin[34], docosahexaenoic acid (DHA)[35], and scallop-derived plasmalogens (PL)[36]), can stimulate the CREB activation (phosphorylation). Unfortunately, these agents have very low bioavailability in the lack of suitable carrier formulations. On the other hand, free drugs predominantly induce short-term transient CREB phosphorylation suggesting the need of frequent drug delivery to achieve a therapeutic effect. The transient phosphorylation of the transcription factor CREB has been determined to last about 20 min in rat cerebellar neurons[28,31]. Therefore, the controlled drug delivery systems for single or multiple drug loading and release can be expected to be more efficient in the management of chronic neurodegeneration effects through promoting sustained neuronal cell survival and proliferation.

The purpose of this study is to determine the impact of designed bioactive LNPs and lipid-peptide nanoassemblies, with different liquid crystalline structural organizations, on the kinetics of CREB activation and neuronal cell recovery from oxidative stress in a PD model in vitro. Vesicular, cubosome, and hexosome types of LNPs were assembled with PUFA-PL, monoolein, and pituitary adenylate cyclase-activating polypeptide (PACAP) as main building blocks (Fig. 1).

**Research hypothesis and nanomedicine-based strategy for neuroregeneration.** PUFA-PL are considered as emergent therapeutics for NDs treatment because of their neuroprotective and anti-apoptotic effects as well as antioxidant properties. PL are glycerophospholipids characterized by a vinyl ether bond at the *sn-1* position of glycerol backbone (Fig. 1a). Plasmalogen deficiency has been indicated as a causative factor for neurodegeneration[37]. In clinical trials, administration of scallop-derived PL has resulted in improved cognitive function of AD patients with mild cognitive impairments after 1 mg of oral daily-dose delivered over several months. Che et al. have suggested that the improved neuroplasticity in the hippocampus of Aβ-treated rats is due to PL-mediated activation of the PI3K-AKT signaling pathway[38] (Fig. 2). The anti-apoptotic effect of PL in damaged neurons has been attributed to activation of both Ras-Raf-MEK-ERK (MAPK) and PI3K-AKT (AKT) pathways[39]. PL may activate the PLC-PKC signaling pathway as well. So far, the development of PL-based therapeutics for the massive clinical trials was hampered because neither purified PL compounds nor the natural sources of PL were readily available in quantities enough for oral drug administration.

We hypothesize that the combined delivery of PL and a neuroprotective peptide by lipid-peptide nanoassemblies may promote CREB activation under neuropathological conditions. PACAP is a natural ligand of the PAC1 G-protein coupled receptor (GPCR) and plays an essential role in neuronal survival[40]. The biological activity of PACAP presents strong interest for the proposed strategy (Fig. 2) as this peptide acts as a neurotrophic factor and may increase the levels of the anti-apoptotic proteins p-AKT, p-ERK1, p-ERK2, and PKC[41,42]. Moreover, nanomedicine may offer new strategies for delivering to the brain and enhancing the therapeutic effects.

The advantages of LNPs with internal liquid crystalline structures[2,5,6,43–47] (e.g., cubosome, hexosome, and spongosome presented in Fig. 1) are the large surface area and stability upon dilution as well as the capacity to simultaneously encapsulate and protect lipo-soluble and water-soluble therapeutic agents (e.g., hydrophobic drugs, bioactive lipids, peptides, proteins, or nucleic acids). Investigations of the relationship between drug delivery and type of liquid crystalline organization of LNPs have established possible sustained-release effects for hydrophobic drugs[10,11].

Here the structures of designed self-assembled PUFA-plasmalogen-loaded LNPs were examined at high resolution by synchrotron small-angle X-ray scattering (SAXS) and cryo-TEM imaging. An in vitro PD model was created by neurotoxin [6-hydroxydopamine (6-OHDA)]- induced oxidative stress using 24h-starved differentiated human neuroblastoma SH-SY5Y cells of a neuronal phenotype[48,49]. The cellular viability was evaluated after treatment with designed nanoparticles. The kinetics of internalization of fluorescently-labeled LNPs and lipid-peptide nanoassemblies was monitored by flow cytometry measurements. We examined whether vesicular or non-lamellar types of LNPs and lipid-peptide nanocarriers, involving PUFA-PL and a neurotrophic peptide (PACAP), may promote neuronal cell regeneration from oxidative stress through a survival mechanism involving CREB phosphorylation. For this purpose, the kinetic changes of the phosphorylated protein (p-CREB, p-AKT, and p-ERK) levels, which regulate the neuronal cell survival, were determined by quantitative ELISA following the treatment of the cellular PD model by LNPs *versus* 6-OHDA-damaged and RA-only treated cells (referred to as RA/FBS(-) control).

## Results

### Structural organization of lipid nanoassemblies involving PUFA-plasmalogens.
Docosahexanenoyl (DHA) plasmenyl

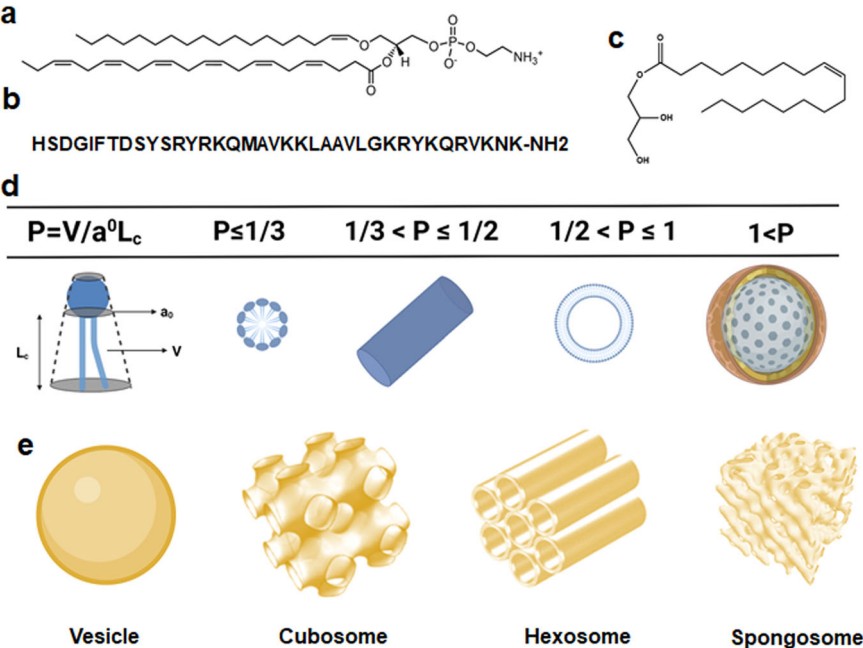

**Fig. 1 Building blocks for the generation of bioactive lipid nanoparticles (LNPs) and lipid-peptide liquid crystalline nanoassemblies. a** Chemical structure of docosahexanenoyl plasmenyl glycerophosphoethanolamine (PL-DHA-PE), which is a PUFA-plasmalogen investigated as an exemplary bioactive vinyl ether phospholipid with neuroprotective and free-radical scavenging properties. **b** Aminoacid sequence of the neurotrophic pituitary adenylate cyclase-activating polypeptide (PACAP), a natural ligand of a G-protein-coupled receptor (GPCR) receptor implicated in neuronal survival. **c** Chemical structure of the lyotropic non-lamellar helper lipid monoolein (MO). **d** Definition of the packing parameter (P) and typical *P* values for amphiphilic mixtures forming self-assembled liquid crystalline nanostructures of lamellar or non-lamellar types. **e** Topologies of vesicle, cubosome, hexosome, and spongosome types of lipid nanoparticles.

glycerophosphoethanolamine PL-DHA-PE (a representative PUFA-plasmalogen shown in Fig. 1a) was encapsulated in LNPs with the purpose of creating liquid crystalline nanoassemblies of non-lamellar (cubosome and hexosome) types. Monoolein was employed as a nonlamellar helper lipid (Fig. 1c) for their stabilization. The molar amount of PL-DHA-PE with regard to monoolein was 15 mol.%. The purified natural scallop-derived plasmalogen was dispersed in the form of LNPs as well. In the present work, the LNPs were sterically stabilized using a PEGylated agent (TPGS-PEG$_{1000}$). Chemical stability of the nanoformulations was achieved by adding small quantities of antioxidants (vitamin E or coenzyme Q$_{10}$) in the lipid mixtures. In the preliminary synchrotron SAXS experiments, we examined the nanostructural organization of LNP formulations of various amphiphilic compositions involving bioactive PL. This allowed us to select stable plasmalogen-based LNPs of vesicular (PL-V), cubosome (PL-C), and hexosome (PL-H) types for further investigation of CREB activation by nanoassemblies in a cellular neurodegeneration model. The stability of the nanoparticles was also confirmed by size measurements (Figure S5). The experimental design considered that the LNP topology and shape are crucial for the interaction with biological membrane barriers. Moreover, the dynamics of these interactions may influence the delivery efficacy of encapsulated drugs to the living cells[50–52].

Synchrotron SAXS patterns characterizing the structure of lipid dispersions containing a natural scallop-derived plasmalogen extract (called scPL70) or a synthetic PUFA-plasmalogen (PL-DHA-PE) are presented in Fig. 3. The Bragg peak position maxima in a SAXS pattern are determined by the periodicity of the structures that scatter the X-rays. Their *q*-vector positions are used for the determination of the structural parameters and the lattice spacing of the generated three-dimensional (3D) liquid crystalline organizations. The absence of Bragg diffraction peaks in the recorded scattering plots implies that no periodic 3D

assemblies are formed upon hydration and dispersion of the lipid phases into nanoparticles.

The scattering curves in Fig. 3a display a maximum centered at $q = 0.101 \text{ Å}^{-1}$ with no detectable Bragg peaks. This fact reveals that the scallop-derived PL scPL70 organize into vesicular structures (PL-V) under the investigated dispersion conditions. The lack of resolved Bragg diffraction peaks indicates that individual vesicular membranes, rather than multilamellar lipid arrangements are formed in the studied scPL70 dispersions (Fig. 3a).

Besides, the PL-DHA-PE plasmalogen carrying a long polyunsaturated fatty acid (C22:6) chain (Fig. 1a) forms stable nonlamellar liquid crystalline phases with the helper lipid monoolein (Fig. 3b, c). Bragg diffraction peaks of nanoassemblies with cubic lattice structures are well resolved at *q*-vector maxima $q_1 = 0.026 \text{ Å}^{-1}$, $q_2 = 0.036 \text{ Å}^{-1}$ and $q_3 = 0.044 \text{ Å}^{-1}$ (see the arrows on Fig. 3b, light blue plot). They were indexed with a set of Bragg peaks spaced in the ratio $\sqrt{2} : \sqrt{4} : \sqrt{6} : \sqrt{8} \dots$, which determined a cubic unit cell lattice parameter $a_{Q(Im3m)} = 37.17 \text{ nm}$ for cubosome type LNPs (PL-C) with primitive *Im3m* cubic space group. The estimated $a_{Q(Im3m)}$ value revealed that the dispersed cubosome particles involve a network of swollen inner aqueous nanochannels. The established nanochannel properties suggest the noteworthy capacity of this non-lamellar type of LNPs for incorporation and encapsulation of guest molecules, e.g., peptides of therapeutic significance such as PACAP.

The sequence of Bragg peak maxima positioned at $q_1 = 0.0526 \text{ Å}^{-1}$, $q_2 = 0.064 \text{ Å}^{-1}$, $q_3 = 0.074 \text{ Å}^{-1}$, $q_4 = 0.091 \text{ Å}^{-1}$, and $q_5 = 0.105 \text{ Å}^{-1}$, and spaced in the ratio $\sqrt{2} : \sqrt{3} : \sqrt{4} : \sqrt{6} : \sqrt{8} : \sqrt{9} \dots$ (see the arrows on Fig. 3b, dark blue plot), determined a cubic unit cell lattice parameter $a_{Q(Pn3m)} = 16.89 \text{ nm}$ for a cubosome type LNPs with a double diamond *Pn3m* inner cubic phase structure (PL-C). The coexisting Bragg peaks with maxima at $q_1 = 0.104 \text{ Å}^{-1}$, $q_2 = 0.181 \text{ Å}^{-1}$ and $q_3 = 0.210 \text{ Å}^{-1}$ (i.e., *q-wave* vector positions spaced in the ratio $1 : \sqrt{3} : \sqrt{4}$) identified the

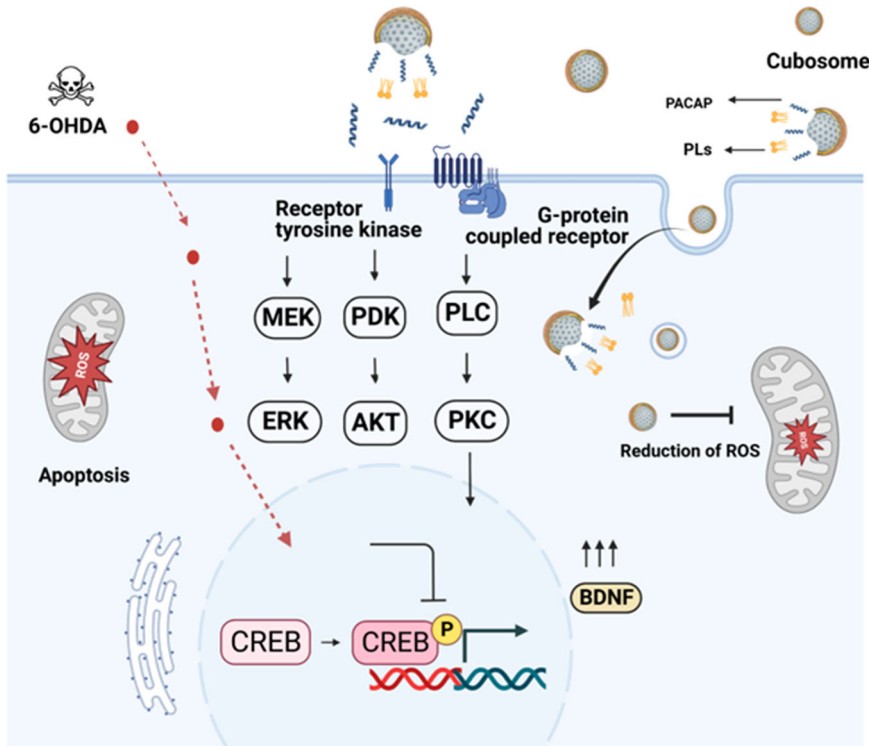

**Fig. 2 Hypothesis of nanomedicine-regulated CREB phosphorylation by lipid-peptide liquid crystalline nanoassemblies of cubosome type in an in vitro Parkinson's disease (PD) model.** Lipid nanoparticles (LNPs), loaded with neuroprotective molecules, may activate CREB through Ras-Raf-MEK-ERK (MAPK), PI3K-AKT (AKT), or PLC-PKC signaling pathways. CREB transcription factor can be phosphorylated by activation of multiple kinases including ERK, AKT, or PKC upon stimulation of a cellular G-protein-coupled receptor (GPCR) or a membrane receptor tyrosine kinase (RTK). In the PD model created in vitro (through cellular SH-SY5Y differentiation by retinoic acid (RA) for 5 days, starvation for 24 h in RA-only medium, and incubation with the neurotoxin 6-OHDA for 30 min), the inhibition of ERK, AKT and PKC pathways may cause mitochondrial dysfunction and trigger cellular apoptosis upon reduced CREB activation. CREB phosphorylation, stimulated by PUFA-plasmalogen-loaded nanoparticles in association with the neurotrophic peptide PACAP, is expected to activate the CREB-BDNF pathway and promote neuroprotection and recovery from oxidative stress-induced neuronal damage. Abbreviations: CREB cyclic AMP (cAMP)-response element-binding protein, AKT Protein kinase B, ERK Extracellular regulated protein kinase, MEK Mitogen-activated protein kinase, Ras Rat sarcoma virus, Raf Rapidly accelerated fibrosarcoma, PLC Phosphoinositol-specific phospholipase C, PKC Protein kinase C, PI3K Phosphatidylinositol-3 kinase, PDK Phosphoinositol-dependent kinase, BDNF brain-derived neurotrophic factor (BDNF), 6-OHDA 6-hydroxydopamine. The signaling pathways were drawn with BioRender (*BioRender.com*).

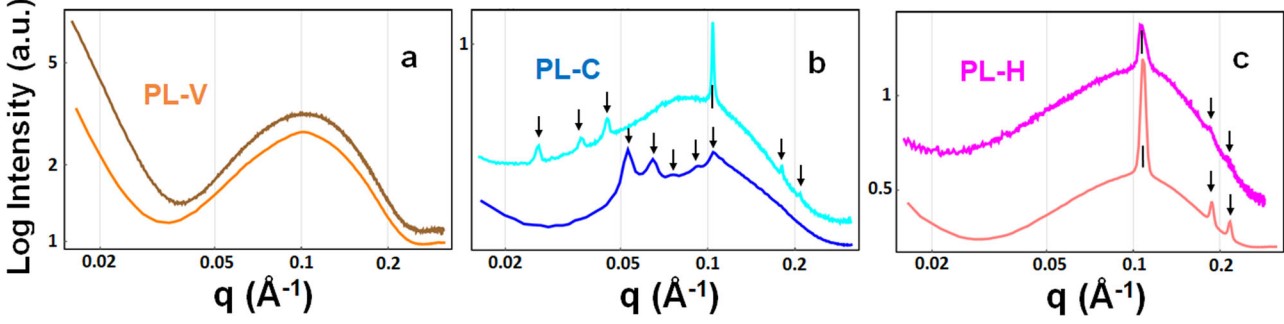

**Fig. 3 Synchrotron small-angle X-ray scattering (SAXS) patterns of plasmalogen-based liquid crystalline LNP dispersions of vesicular (PL-V), cubosome (PL-C), and hexosome (PL-H) types. a** Scallop-derived PL extract (called scPL70) forms vesicular membrane structures (PL-V) obtained upon dispersion of the lamellar liquid crystalline phase by PEGylated agent TPGS-PEG$_{1000}$. **b, c** Cubosome (PL-C) and hexosome (PL-H) types of LNPs are obtained by embedding synthetic docosahexaneoyl (DHA) glycerophospholipid PL-DHA-PE in self-assembles mixtures with hydrated monoolein at MO/PL-DHA-PE molar ratio 85/15 mol/mol (see Table S1 for the compositions of scPL70- and PUFA-plasmalogen-based formulations). Aqueous phase: $1 \times 10^{-2}$ M phosphate buffer containing butylated hydroxytoluene (BHT). The temperature is 22 °C. The arrows indicate the positions of the Bragg peaks discussed in the text.

presence of a fraction of hexosome nanoparticles with a unit cell lattice parameter $a_{HII} = 6.97$ nm. The first peak of the inverted hexagonal structure overlaps with the Bragg peak of cubic *Pn3m* structure at $q = 0.105$ Å$^{-1}$. The obtained structural results corroborate with the organization of the nanoassemblies

embedding docosapentaneoyl (DPA) plasmenyl glycerophosphoethanolamine (PL-DPA-PE), a plasmalogen derivative with a C22:5(n-6) PUFA chain[47]. In mixtures with the helper lipid MO, both PL-DHA-PE and PL-DPA-PE-based nanoassemblies form non-lamellar inverted cubic structures (cubosomes) dispersed in a

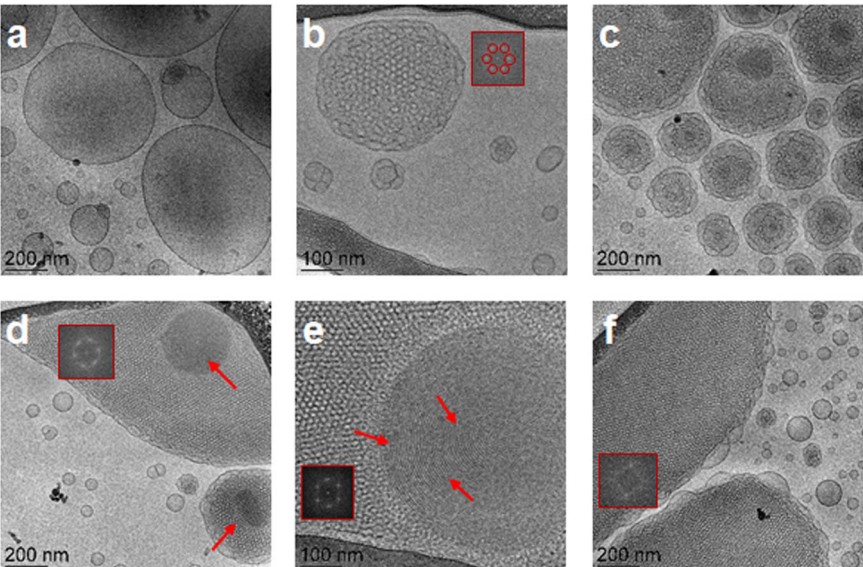

**Fig. 4 Cryo-TEM images of liquid crystalline lipid nanoparticles of vesicular (PL-V), cubosome (PL-C), and hexosome (PL-H) types of LNPs obtained by dispersion of PUFA-plasmalogen-based liquid crystalline lipid mixtures in excess aqueous medium.** The PL-V (**a**), PL-C (**b–d**), and PL-H (**e, f**) compositions correspond to the LNPs studied by SAXS in Fig. 3 (PL-V: Fig. 3a, orange plot; PL-C: Fig. 3b, dark blue plot; and PL-H: Fig. 3c, pink plot). At the investigated submicromolar concentration, the peptide PACAP did not influence the liquid crystalline organization of the PL-C nanoassemblies. Insets: Fourier Transform patterns revealing the formation of cubic and inverted hexagonal lattices. The arrows indicate the striations of inverted hexagonal lipid packing.

coexistence with a fraction of inverted hexagonal phase structures (hexosomes).

Self-assembled LNPs of the hexosome-type (PL-H) were identified as a major population in PUFA-plasmalogen-loaded nanoformulations prepared without $Q_{10}$ in the mixture (Table S1). The Bragg peak positions at $q_1 = 0.108$ Å$^{-1}$, $q_2 = 0.188$ Å$^{-1}$, and $q_3 = 0.217$ Å$^{-1}$ in the SAXS pattern shown in Fig. 3c (pale pink plot) determined a periodic inverted hexagonal structure with a lattice parameter $a_{HII} = 6.71$ nm. The scattering pattern in Fig. 3c (dark pink plot) is characterized by slightly lower intensity, but similar Bragg peak positions, which yield a lattice parameter $a_{HII} = 6.71$ nm for the inner structure of hexosome-type of LNPs (PL-H). The smaller $a_{HII}$ lattice parameter as compared to the cubic $a_Q$ lattice parameter implies that the density of the lipid building blocks (lipid tubes or bicontinuous bilayer membranes) is higher in the supramolecular structure of the hexosome PL-V nanocarriers as compared to the cubosome PL-C ones. This structural feature may influence the bioactive compound release and the interaction of the two types of LNPs with the biological membrane interfaces.

The performed SAXS experiments found that the addition of PACAP in the formulations, at peptide concentrations corresponding to therapeutic conditions (submicromolar concentration), does not modify the structural organization of self-assembled plasmalogen-based nanoassemblies of cubosome (PL-C), hexosome (PL-H), or vesicular (PL-V) types.

The structural features of the plasmalogen-based LNP formulations established by cryo-TEM imaging (Fig. 4) corroborate with the results of the synchrotron SAXS analysis (Fig. 3). Figure 4a presents a typical image of vesicular structures (PL-V), which are formed by self-assembly and dispersion of the natural scallop-derived plasmalogen extract (scPL70). They represent single bilayers, which do not display a long-range crystalline order of multilamellar bilayer organization. This result is in agreement with the lack of Bragg diffraction peaks at low $q$-vectors in the SAXS pattern of the PL-V samples (Fig. 3a, orange plot).

When adding the helper lipid MO to PUFA-plasmalogen/water systems, the vesicular membranes topologies are transformed into

nonlamellar structures (Fig. 4b, c). The cryo-TEM images in Fig. 4(b, c, d) correspond to the sample PL-C prepared by dispersion of the PL-DHA-PE-based lipid mixture (SAXS pattern in Fig. 3b, dark blue plot), to which PACAP was added at a therapeutic submicromolar concentration. The formation of cubosome type LNPs was evidenced and supported by the cryo-TEM results as well. The topology of the inner network of aqueous channels represents the structural organization of multicomponent self-assembled lipid/peptide mixture. The cryo-TEM images reveal that the nonlamellar structures involve domains resulting from the packing of different amphiphilic ingredients in the nanoassemblies. The cubosome LNPs with a regular periodic inner organization (Fig. 4b) coexist with cubosomes with large channels (Fig. 4c) as well as with cubosomes involving domains with more dense packing (Fig. 4c).

The cryo-TEM images in Fig. 4(e, f) correspond to the hexosome-type LNP sample (PL-H). They indicate that the LNPs displaying Bragg peaks of an inverted hexagonal structure (dark pink plot in the SAXS pattern on Fig. 3c) yield topologies with periodic inner organization with more dense inner packing than that of cubosomes (Fig. 4e, f). The striations observed in Fig. 4e represent the packing of lipid tubes into inverted hexagonal phase domains and confirm the formation of hexosome-type structures (PL-H).

The performed Fourier Transform analysis of cryo-TEM images yielded cubic and inverted hexagonal phase organization (presented as insets in panels b, d, e and f of Fig. 4). The FFT patterns in Fig. 4(b, c, d) reveal *Pn3m* cubic lattice formation, whereas inverted hexagonal domain structures are identified by the FFT patterns in Fig. 4(e, f). The obtained cryo-TEM data revealed that the packing requirements of the PUFA-plasmalogen lipid molecules, such as PL-DHA-PE, are different from those of the helper matrix lipid and favor nanodomain formation.

**In vitro PD model involving oxidative stress.** The in vitro PD model was created by inducing oxidative stress and neurotoxicity with 6-OHDA in human SH-SY5Y cells, which were

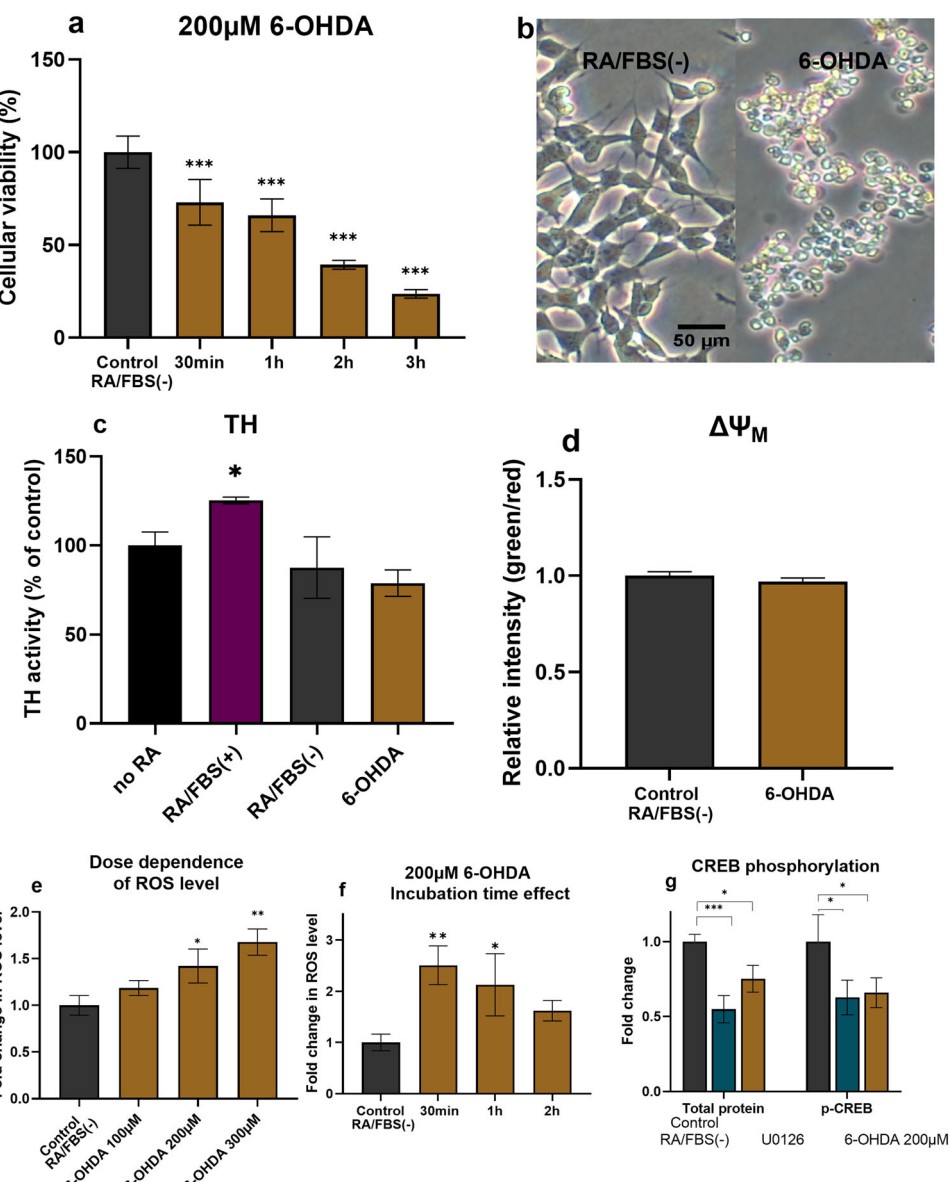

**Fig. 5 In vitro PD model induced by 6-OHDA with differentiated SH-SY5Y cells, which were starved for 24 h in RA-only medium. a** MTT results acquired after cells exposure to 200 µM 6-OHDA for 30 min, 1 h, 2 h, and 3 h in RA/FBS(-) medium. The data is presented as a ratio of mean absorbance at 570 nm in the treated group ($n = 6$) compared to the control RA/FBS(-) group ($n = 6$). **b** Morphological analysis conducted after cells were exposed to 200 µM 6-OHDA for 30 min in RA/FBS(-) medium. **c** Tyrosine hydroxylase expression assayed by ELISA with no RA: non-differentiated SH-SY5Y cells, RA/FBS(+): RA-differentiated cells. RA/FBS(-):RA-differentiated cells with 24 h FBS deficient medium incubation. 6-OHDA: RA/FBS(-) with 200 µM 6-OHDA for 30 min. **d** Mitochondrial membrane potential assay conducted with RA/FBS(-) cells exposed to 200 µM 6-OHDA for 30 min. The results are presented as a ratio of the mean green fluorescence intensity (FL1) divided by mean red fluorescent intensity (FL2) in the treated group ($n = 3$) compared to RA-control group ($n = 3$). **e** Reactive oxygen species (ROS) assay conducted with different concentrations (100 µM, 200 µM, 300 µM) of 6-OHDA for 30 min. The data are presented as a ratio of the mean fluorescent (FL1) intensity in the treated group ($n = 3$) compared to the RA/FBS(-) control group ($n = 3$). **f** Reactive oxygen species assay conducted with 200 µM of 6-OHDA for different incubation times (30 min, 1 h and 2 h). **g** Quantitative ELISA assay used to determine CREB phosphorylation after 200 µM 6-OHDA incubation for 30 min. The data are presented as the folded change of CREB phosphorylated protein (ng) per total protein (ng) in the treated group ($n = 3$) compared to the RA/FBS(-)control group ($n = 3$). $n$ indicates the number of replicates per group and the error bars represent the standard deviation. The Dunnett test was used for multiple comparisons, and the Student $t$ test was employed for comparing two groups, both using the Prism software. *$P \leq 0.05$, **$P \leq 0.01$, and ***$P \leq 0.001$.

differentiated by retinoic acid (RA) for 5 days and then starved for 24 h in RA-only medium RA/FBS(-). The qualitative and quantitative characteristics of the studied PD model are presented in Fig. 5(a–g) (for source data see Supplementary Data 1 file). The deprivation of serum (24 h) from the culture medium of the RA-differentiated SH-SY5Y cells was done based on published findings that it increases the cellular susceptibility to the environment

and the applied treatment[53,54]. Serum starvation facilitates the induction of neurite formation and prevents excessive growth of S-type SH-SY5Y cells. Moreover, SH-SY5Y cells when subjected to nutrient deprivation, display a higher responsiveness to 6-OHDA that induces apoptosis and cell death. The morphologies of differentiated SH-SY5Y cells, both before and after 24 h starvation, are shown in Figure S1 and Figure S2 of

the Supplementary Information. The capacity of 6-OHDA to induce the degeneration of dopaminergic neuron has been evidenced with animal PD models[49].

RA treatment enhanced the dopaminergic properties of the SH-SY5Y cells (Fig. 5c). Tyrosine hydroxylase (TH) expression level was used as an indicator of the degree to which cells have successfully differentiated into dopaminergic neuronal cells[55]. This enzyme is implicated in the conversion of tyrosine to dopamine and plays a key role in the production of important neurotransmitters (e.g., dopamine, norepinephrine, and epinephrine)[56]. The TH activity levels presented in Fig. 5c indicate a significant increase in TH expression in the RA-differentiated SH-SY5Y cells as compared to either the non-differentiated cells or the group treated with the neurotoxin 6-OHDA. The data in Fig. 5c indeed show that exposure to 6-OHDA reduces the TH activity.

The cellular viability results indicated that 30 min of incubation with 200 μM 6-OHDA decreased the neuronal cell viability up to 70% as compared to the RA/FBS(-) control cells (Fig. 5a). The cellular viability considerably decreased as 6-OHDA incubation time increased (Fig. 5a). Moreover, the neurotoxic damages induced by 6-OHDA were dose-dependent. In this work, the in vitro PD model was generated with 30-min cellular exposure to freshly prepared 6-OHDA solution to avoid other mechanisms that might be induced by degradation products over long exposure times. The microscopic analysis of cellular damage under oxidative stress conditions (Fig. 5b) demonstrated that 30-min exposure to 200 μM 6-OHDA causes neuronal cells to lose their elongated shape and to become rounded. Such morphological changes suggest an early stage of apoptosis in response to 6-OHDA-induced stress.

The mitochondrial membrane potential changes were monitored by flow cytometry (Fig. 5d) using the fluorescence (green/red) intensity ratio of a dye probe present in an aggregated state (red fluorescence in intact mitochondria of healthy cells) or in a monomeric state (green fluorescence when the dye leaks from the mitochondria of apoptotic cells). An increasing green/red intensity ratio is related to abnormal membrane potential and mitochondrial damage. Figure 5d demonstrates that the mitochondrial membrane depolarization by 6-OHDA slightly changes over 30 min compared to the control group but does not reach statistical significance ($P = 0.13$).

Reactive oxygen species (ROS) were measured at different 6-OHDA concentrations and time points. The results in Fig. 5e show that the ROS levels increase with the progressive increase in the neurotoxin concentration. This reflected the dose-dependent ROS response to 6-OHDA. Figure 5f indicates that the measured ROS levels are maximal at 30-min cellular exposure to 200 μM 6-OHDA and begin to drop with increasing incubation time. The reduction in ROS levels with time is likely due to the progression of cellular apoptosis process from its early stage to later stages.

We further investigated whether 6-OHDA-induced neuronal death mechanism may involve CREB signaling pathway, which plays an essential role in cellular proliferation. Quantitative ELISA was utilized to assess the phosphorylated CREB levels. As a positive control, we employed U0126, which is a specific inhibitor of MEK1 and MEK2 that reduces CREB expression[57]. The results in Fig. 5g show that 6-OHDA significantly reduced CREB phosphorylation, similarly to the effect of the positive control U0126. The observed down-regulation of p-CREB confirmed that one of the toxic mechanisms of 6-OHDA on the dopaminergic neuronal cells is related to CREB pathway[58].

The obtained results revealed that 6-OHDA-treated neuronal-type SH-SY5Y cells exhibit the following characteristics in the created in vitro PD model: (i) metabolic damage (as proven by MTT assay), (ii) a high level of ROS, and (iii) reduced CREB

activation. Therefore, this model represents some of the fundamental neurodegenerative conditions for dopaminergic neuronal cells, particularly in the context of Parkinson's pathology. Along the line, we further investigate the capacity of LNP formulations in restoring neuronal damages through activation of CREB signaling in this in vitro PD model.

**Regenerative effects of PUFA-plasmalogen-loaded lipid nanoparticles on the in vitro PD model.** Notable increase in neurite extension and a significant decrease in cell body condensation were achieved after 24 h treatment of the cellular PD model by PUFA-plasmalogen-based LNPs (Fig. 6a). The microscopy analysis of neuronal morphology demonstrated that the differentiated SH-SY5Y cells can recover from the early-stage apoptotic state (induced by 6-OHDA exposure) to their normal shape after 24 h treatment by PL-H LNPs. The performed MTT test (Fig. 6b) indicated that the investigated LNPs are not toxic. Interestingly, the cellular viability after treatment by hexosomes (PL-H) was greater than the control group RA/FBS(-). This fact evidenced that PL-H LNPs have a positive effect on neurite growth. Moreover, the results indicated that the nonlamellar plasmalogen-loaded LNPs (PL-H and PL-C) are more efficient at repairing the damages caused by 6-OHDA as compared to blank LNPs or the cell culture medium alone.

The in vitro PD cellular model was incubated with 10 μM plasmalogen-loaded LNPs (PL-H and PL-C) for 24 h to evaluate the regenerative capacity against the oxidative stress damage. Afterwards, cell viability, ROS levels, and changes in mitochondrial membrane potential were measured (Fig. 6c, d, e). The MTT assay revealed that the treatment by cubosome PL-C and hexosome PL-H particles considerably increases cellular viability as compared to 6-OHDA group (Fig. 6c). The recovery effect was supported by the microscopic results as well (Fig. 6a, right panel). Figure 6d shows that the ROS levels significantly decreased in cubosome PL-C and hexosome PL-H LNP-treated groups as compared to the 6-OHDA group.

Additionally, the mitochondrial membrane potential changes, evaluated by the ratio of green (apoptotic cells) / red fluorescence (normal cells) intensities, showed a decreasing trend in the LNP-treated groups. This indicated that incubation with cubosomes (PL-C) or hexosomes (PL-H) minimizes the mitochondrial membrane injury and regenerates the differeciated-SH-SY5Y cells from oxidative damage. The obtained findings imply that 6-OHDA (200 μM, 30 min)-induced damages are reversible upon treatment by plasmalogen-based nanoformulations. The neuronal cell regeneration appears to be enabled by LNP-mediated reduction of oxidative stress and inhibition of the mitochondrial membrane damages.

Overall, the results presented in Fig. 6 indicated that PUFA-plasmalogen-based hexosome (PL-H) and cubosome (PL-C) LNPs could regenerate differentiated SH-SY5Y cells after short-term 6-OHDA-induced damage.

**Kinetic changes of CREB phosphorylation modulated by PUFA-plasmalogen-loaded lipid nanoparticles in a cellular PD model.** In the lack of previous reports, we suggested that the established 6-OHDA-induced apoptotic-like morphological alterations and decreased cellular viability may be related to the downregulation of p-CREB (a crucial factor for cellular differentiation and growth). According to the literature, CREB phosphorylation/activation lasts around 20 min after free drug administration[59]. For healthy cells, sustained CREB phosphorylation is not needed and should be avoided. For the pathological conditions of reduced CREB phosphorylation observed here, we addressed the question of whether CREB activation can be

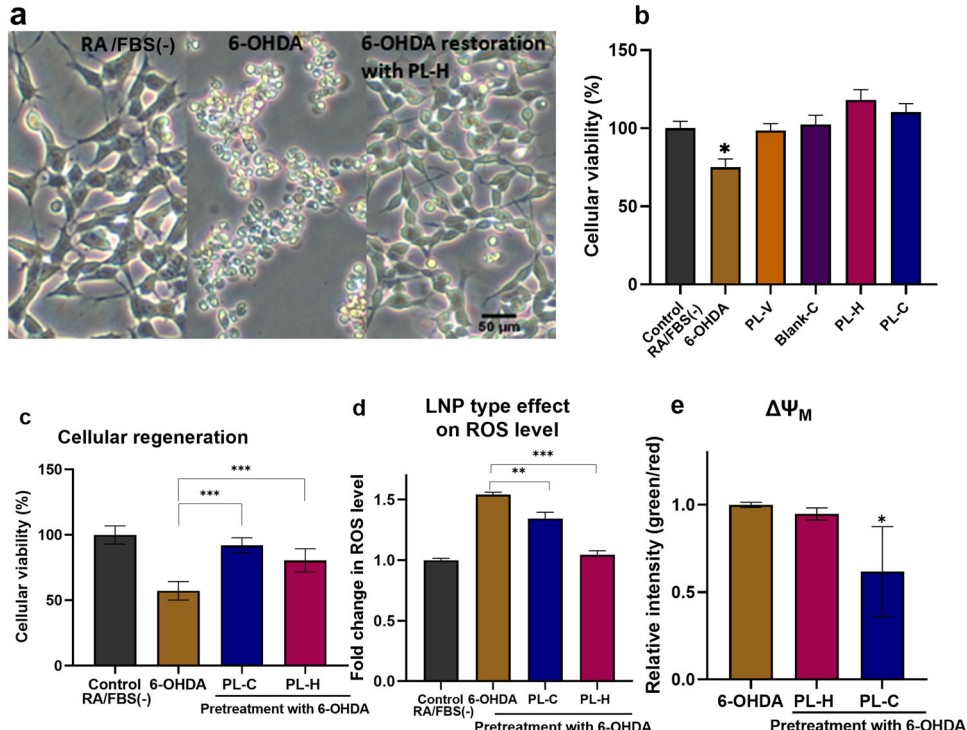

**Fig. 6 Regenerative effects of PUFA-plasmalogen-based particles on the in vitro PD model. a** Morphological analysis of differeciated-SH-SY5Y cells after exposure to 200 µM 6-OHDA for 30 min. Images of three groups are compared: RA/FBS(-) as a control sample, 6-OHDA (in vitro PD model using 30 min 6-OHDA exposure after incubation in serum-free medium for 24 h), and regeneration of the PD model mediated by PL-H (10 µM LNPs, 24 h incubation). **b** MTT results obtained following a 30 min exposure to 6-OHDA or 24 h exposure to a blank cubosome carrier and plasmalogen-loaded LNPs (PL-H and PL-C). The data are presented as a mean ratio of viable cells in the treated group ($n = 6$) compared to the RA/FBS(-) control group ($n = 6$). **c** Cellular viability assay in neuroprotective experiments with plasmalogen-loaded cubosome and hexosome LNPs. The MTT results were acquired after the cells were exposed to 200 µM 6-OHDA for 30 min in RA/FBS(-) medium following by 24 h incubation with PL-C and PL-H LNPs. **d** ROS level changes in the in vitro PD model after 24 h treatment by plasmalogen-loaded PL-C and PL-H LNPs. The data are presented as a ratio of FL1 mean fluorescent intensity in the treated group ($n = 3$) compared to RA/FBS(-) control group ($n = 3$). **e** Mitochondrial membrane potential changes in the in vitro PD model after 24 h recovery by cubosome PL-C and hexosome PL-H LNPs. The data is presented as a ratio of green (FL1) mean fluorescent intensity to red (FL2) mean fluorescent intensity in the treated group ($n = 3$) as compared to the same ratio in the RA/FBS(-) control group ($n = 3$). All the experiments were conducted with SH-SY5Y cells after cellular differentiation by 10 µM RA for 5 days, followed by 24 h of incubation in a serum-free medium. $n$ indicates the number of replicates per group and the error bars represent the standard deviation. Differences were evaluated by one-way ANOVA. *$P \le 0.05$, **$P \le 0.01$, and ***$P \le 0.001$.

enhanced or prolonged by cubosome (PL-C) and hexosome (PL-H) LNPs, which may have longer residence time at the biological barriers as compared to conventional vesicles. We supposed that the LNPs might protect the bioactive plasmalogen from oxidation or degradation and provide longer-lasting regenerative effects in the PD model. The results presented in Figures S3 and S4 in the Supplementary Information indicate that p-CREB is significantly reduced in the in vitro PD model (6-OHDA) as compared to the control RA/FBS(-) and the LNP-treated groups.

Using quantitative ELISA, we examined the phosphorylation of CREB, AKT and ERK proteins as crucial factors for the LNP-mediated neuronal regeneration in the investigated in vitro PD model. Figure 7 presents the variations in the p-CREB (Fig. 7a, d, g), p-AKT (Fig. 7b, e, h), and p-ERK (Fig. 7c, f, i) levels over time following the treatment of cellular PD model by vesicular (PL-V), hexosome (PL-H), or cubosome (PL-C) types of LNPs, respectively. The time dependencies in Fig. 7 indicate that the vesicular assemblies (PL-V) have a shorter-term efficacy on protein phosphorylation (data given on the first row of Fig. 7) with regards to the non-lamellar types of PUFA-plasmalogen-loaded hexosome (PL-H) and cubosome (PL-C) LNPs (data presented in the second and third rows of Fig. 7). In the PL-V group, the phosphorylated protein levels reached their maximum

values 6 h after the LNPs administration, while these levels decreased significantly at 24 h time point (Fig. 7a, b, c). The observed kinetic changes are essential. The maximal p-CREB levels found 6 h after the onset of LNP treatment likely reflects the convolution of the time needed for internalization of the LNPs, the time for activation of the plasmalogen receptors by released PL molecules, and the duration of subsequent intracellular signaling pathways, which regulate CREB phosphorylation.

For the cubosome PL-C and hexosome PL-H groups, the p-CREB levels were consistently elevated, particularly in the PL-H group. This suggested that nonlamellar LNPs may facilitate the internalization of encapsulated PUFA-plasmalogen or its interaction with its receptors and transporters (Fig. 2). The achieved steady p-CREB levels demonstrated the sustained effect of cubosome LNPs (Fig. 7g). After 24 h incubation with PL-C LNPs, CREB phosphorylation levels were nearly equal to those at 6 h. The decrease in the p-CREB levels after 24 h LNP incubation in the other groups may be due to degradation of the released PL. It can be assumed that PL-C and PL-H provide better protection of plasmalogen against oxidation or degradation in comparison to PL-V. The AKT and ERK phosphorylation processes were also prolonged by PL-H (Fig. 7e, f) and PL-C (Fig. 7h, i) LNPs with regard to PL-V group (Fig. 7b, c).

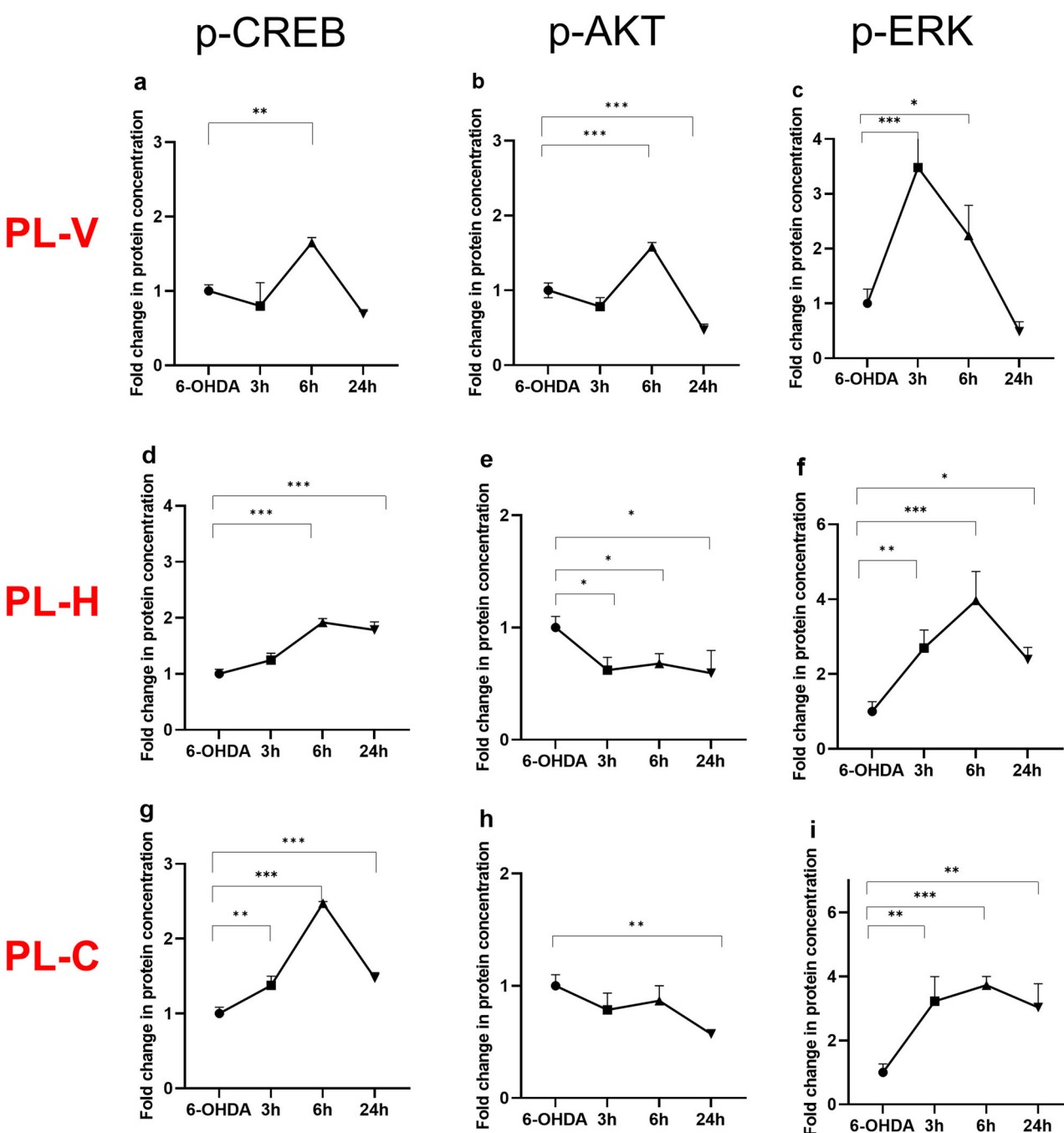

**Fig. 7 Kinetic changes of phosphorylation of CREB, AKT, and ERK proteins following LNP-treatment of the in vitro PD model.** The time changes in protein phosphorylation were measured using quantitative ELSA assays. The nanoformulations were of the same total lipid concentration (10 µM). The incubation times of vesicular PL-V (**a–c**), hexosome PL-H (**d–f**), and cubosome PL-C (**g–i**) LNPs were 0, 3 h, 6 h, and 24 h. The fold changes are presented as a ratio of the quantities of phosphorylated proteins (ng) per total protein (ng) in the treated group ($n = 3$) and the same ratio in the 6-OHDA group ($n = 3$). $n$ indicates the number of replicates per group and the error bars represent the standard deviation. *$P \leq 0.05$, **$P \leq 0.01$, and ***$P \leq 0.001$.

**Kinetic changes of CREB phosphorylation modulated by PUFA-plasmalogen-loaded LNPs in nanoassemblies with a pituitary adenylate cyclase-activating polypeptide (PACAP).** We created lipid-peptide nanoassemblies using plasmalogen-based LNPs (PL-C) and neurotrophic peptide PACAP. Figure 8(a–c) compares the internalization kinetics of fluorescently labeled LNPs, PACAP labeled by TAMRA dye, and LNP-PACAP(TAMRA) nanoassemblies. The considerably higher

fluorescence intensity ratios in Fig. 8b, c with regard to the LNPs without PACAP (Fig. 8a) confirmed the cell-penetrating properties of the peptide and its capacity to facilitate LNPs internalization. PACAP can exert synergistic biological effects in the activation of neurotrophic intracellular signaling pathways. The kinetics of CREB activation depends on many factors, e.g., the residence time of the LNPs at the cellular membrane (Fig. 2), their interaction with surface-exposed membrane protein

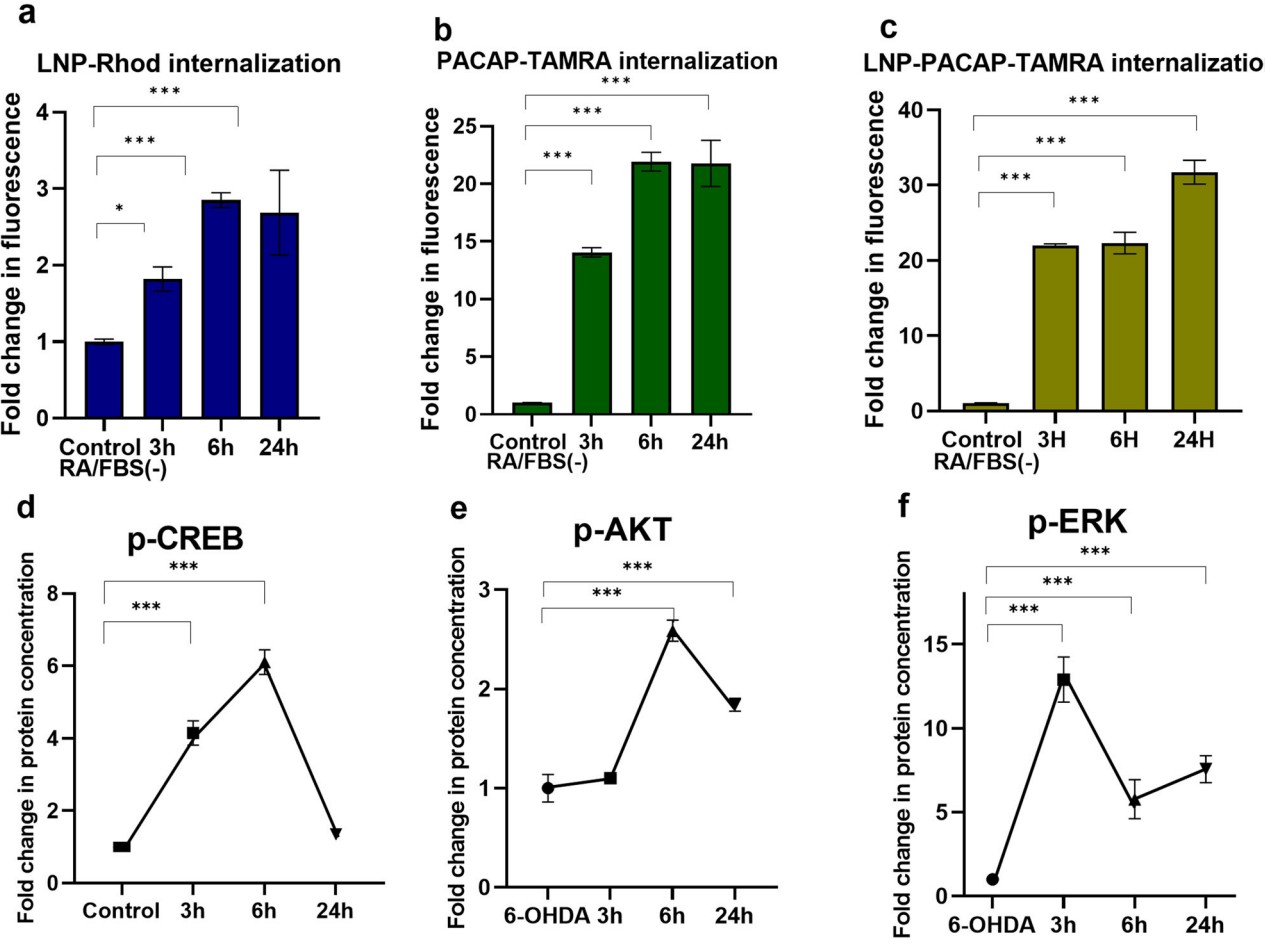

**Fig. 8 Internalization kinetics (top row) and kinetic changes of CREB, AKT, and ERK phosphorylation in an in vitro PD model treated by nanoassemblies of PL-C and PACAP (bottom row). a** LNP cellular uptake measured by flow cytometry: Internalization of Rhodamine R18-labeled MO cubosomes (10 µM) in differentiated SH-SY5Y cells after 24 h of incubation (determined by FL2 fluorescence intensity). **b** Cellular uptake of PACAP: Internalization of PACAP-TAMRA (5 µM) in differentiated SH-SY5Y cells at different incubation times of 3 h, 6 h and 24 h (determined by FL2 fluorescence intensity). **c** Cellular uptake of LNP-PACAP: Internalization of LNP-PACAP-TAMRA by differentiated SH-SY5Y cells from 5 µM PACAP-TAMRA/10 µM LNPs mixture at different time points 3 h, 6 h and 24 h. The data of (**a–c**) was presented as a ratio of the mean intensity of FL2 in the treated group ($n = 3$) and the mean intensity of FL2 in the RA/FBS(-) control group. **d** Kinetic changes of CREB phosphorylation in the PD model. The ELISA assay was performed with plasmalogen-loaded PL-C LNPs, which were dispersed at total lipid concentration of 10 µM in the presence of 5 µM PACAP peptide. The different incubation times were 0, 3 h, 6 h, and 24 h following 30 min 6-OHDA exposure. **e** Kinetics changes of phospho-AKT expression levels in the treated PD model. ELISA assay was performed with plasmalogen-loaded nanoparticles PL-C, which were dispersed at same total lipid concentration of 10 µM in the presence of 5 µM PACAP. **f** Kinetic changes of phospho-ERK expression in the treated PD model measured by ELISA assay with plasmalogen-loaded nanoparticles PL-C at total lipid concentration of 10 µM in the presence of 5 µM PACAP peptide. The data of (**d–f**) is presented as a ratio of the quantity of p-CREB, p-AKT or p-ERK expression (ng) per total protein (ng) in the treated group ($n = 3$) and the quantity of p-CREB expression (ng) per total protein (ng) in the 6-OHDA group ($n = 3$). $n$ indicates the number of replicates per group and the error bars represent the standard deviation. $^*P \leq 0.05$, $^{**}P \leq 0.01$, and $^{***}P \leq 0.001$.

receptors, as well as LNP internalization, which can be accelerated by the associated cell-penetrating peptides (such as the employed here PACAP).

The results for CREB, AKT and ERK-protein phosphorylation by cubosome lipid-peptide nanoassemblies are presented in Fig. 8(d–f). The data in Fig. 8d reveal significantly high levels of activated CREB after 3 h treatment by LNP-PACAP nanoassemblies as compared to the 6-OHDA group. The LNP-peptide treated group exhibited nearly a four-fold increase in CREB phosphorylation compared to the diseased state. These results suggest that PACAP-LNP nanoassemblies can enhance CREB phosphorylation at shorter times (Fig. 8d) and cellular regeneration as compared to LNPs alone (Fig. 7g). The CREB activation decreased after 24 h exposure to nanoassemblies. The kinetic changes of p-AKT levels over time (Fig. 8e) displayed a different

trend from the plasmalogen-LNPs treated group (Fig. 7h). Besides, the p-ERK levels reached their maximal values after 3 h treatment (Fig. 8f) instead of 6 h (Fig. 7i). Although the multiple mechanisms by which PACAP modulates p-CREB kinetics require further investigation, our study demonstrates the ability to control CREB phosphorylation kinetics by lipid-peptide nanoassemblies under pathological conditions.

All together, these data show that the investigated lipid-peptide liquid crystalline nanoassemblies can regulate the CREB phosphorylation in a sustainable way under pathological conditions. PUFA-plasmalogen-loaded LNPs may control the phosphorylation of CREB through the AKT and ERK signaling pathways. Reduction in oxidative stress is another primary mechanism contributing to the neuroprotective effect of PUFA-plasmalogen-loaded LNPs.

## Discussion

Protein transcription factor phosphorylation plays a key role in cell signaling and disease mechanisms but has been little studied in intervention therapies using nanomedicines. Whereas several reports have focused on the toxicity of different kinds of nanoparticles and the induction of cell death pathways[52,60,61], the nanomedicine-controlled CREB phosphorylation appears to be an emergent therapeutic option towards neuroregeneration[8]. This strategy accounts for the fact that neurotoxicity, underlying the neurological disorders, is associated with decreased CREB phosphorylation and hampered expression of neurotrophic proteins, which are responsible for the neuronal cell survival, neuronal plasticity, and function.

Knowledge about the mechanisms by which nanomedicines (derived from PUFA-plasmalogen-loaded LNPs) may control the spatio-temporal dynamics of CREB phosphorylation will potentially lead to more effective treatments of NDs and neurological sequelae of the long COVID syndrome. LNPs prepared by self-assembly are gaining increasing interest for drug delivery due to their biocompatible nature and capacity of cargo protection[2,5,50,51,62,63]. We created cubosome (PL-C), hexosome (PL-H), and vesicular (PL-V) types of liquid crystalline LNPs as a more advanced generation of nanocarriers for improvement of the PL administration. Such drug delivery systems are needed for therapeutic uses of PL in nanomedicine aiming at recovery from oxidative stress-induced neurodegeneration. The obtained high-resolution SAXS and cryo-TEM results (Figs. 3, 4) revealed important information of the structural complexity, inherent of pharmaceutically relevant multicomponent compositions, and the occurrence of coexisting liquid crystalline nanoassemblies.

For PUFA-plasmalogen bioactive lipid species, we achieved the nanomedicine-controlled activation of CREB in a cellular PD model. The studied neuroprotective LNPs and neurotrophic lipid-peptide nanoassemblies appear to activate both AKT and ERK signaling pathways, for which CREB is a downstream target (Fig. 7). The obtained data (Figs. 7, 8) demonstrated that PL-C, PL-H, and PL-V LNPs alone or in combination with a neurotrophic peptide, such as PACAP, influence the phosphorylation of AKT and ERK. Enhanced CREB activation was observed in the presence of the neurotrophic peptide PACAP, which displays cell-penetrating properties and accelerates the kinetics of LNP uptake (Fig. 8). Upon PUFA-plasmalogen delivery, cubosome (PL-C) liquid crystalline nanocarriers followed by hexosomes (PL-H) achieved CREB phosphorylation during an extended period (24 h) (Fig. 7). Thus, the non-lamellar lipid assemblies displayed a remarkable capacity of sustained CREB activation.

Figure 9 summarizes the time kinetic changes of the phosphorylation of CREB as determined by quantitative ELISA following the treatment of the cellular PD model by LNPs. The main result for this protein, which regulates the neuron cell survival, is that cubosome liquid crystalline LNPs, formulated with PUFA-PL, can significantly prolong the CREB activation up to 24 h. Owing to its cell-penetrating properties, the neurotrophic peptide PACAP accelerates the internalization kinetics of lipid-peptide assemblies and at shorter time (e.g., 3 h) promotes enhanced CREB activation with regard to pure LNPs (lacking peptide molecules). For instance, the fold change in the p-CREB levels for 3 h treatment by cubosomes PL-C is 1.4, whereas it augments to nearly 4 in the (PL-C + PACAP) group.

In a similar way, cubosome and hexosome LNPs can be used as a platform for encapsulation and delivery of other natural neuroprotective compounds, which may also stimulate CREB phosphorylation (e.g., curcumin, PUFA conjugates, etc.) under neurodegenerative conditions. The primary aim is to achieve efficient regeneration from oxidative stress-induced neuronal damages through sustained CREB phosphorylation. An additional advantage of the investigated lipid-peptide nanosystem (constituted of mucoadhesive amphiphiles and cell-penetrating therapeutics) is that it can be delivered locally, for example by intranasal administration. This prospective platform should aim at nanomedicine-triggered neuronal survival pathways upon non-invasive drug delivery by LNPs.

The limitations of this study are associated with the selected in vitro disease model of neurodegeneration. In the absence of previously published information concerning the spatio-temporal kinetics of CREB activation by lipid-peptide nanoassemblies and considering that CREB phosphorylation typically lasts about 20 min for signaling pathway activation by free drugs, we developed our in vitro neurodegeneration model through short-term exposure to the neurotoxin 6-OHDA. To enhance the responsiveness of the RA-differentiated SH-SY5Y cells to short-term damage induced by 6-OHDA, we exposed the cells to 24 h of pre-treatment in serum-depleted medium, specifically, 24 h of starvation in an RA-only medium. The obtained data about LNP-mediated neuronal cell recovery from oxidative stress indicate that cellular viability is higher in the PL-H group than in the control RA/FBS(-) sample, as cellular viability is also higher in the PL-H group than in the 6-OHDA group. Thus, PL-H may be able to inhibit 6-OHDA-induced apoptosis as well as general apoptosis, likely because LNPs can serve as a nutrient source for the cells. We have previously reported that PUFA-based LNPs (e.g., cubosomes loaded with DHA) promote the cellular survival of 24h-starved RA-differentiated SH-SY5Y cells and increase their viability[64]. The present work confirms that plasmalogen-based LNPs may also favor recovery from short-term oxidative damage caused by neurotoxin 6-OHDA in addition to recovery from starvation that leads to cellular apoptosis.

The exact signaling pathway and dynamics of CREB activation by plasmalogen species in damaged neuronal cells is still not well understood. PUFA-PL have free-radical scavenging and anti-oxidant properties. In addition, they have been suggested to activate GPCRs in the cellular lipid membranes (Fig. 2), thus acting as hormones[65]. Therefore, both membrane receptor activation and intracellular neurotrophic signaling may be evoked to explain the effect of PL-based nanoassemblies on CREB phosphorylation. Whereas the present study focused on the capacity of PUFA-plasmalogen-based nanoformulations to prolong the kinetics of CREB phosphorylation beyond 1 h, PUFA-PL are bioactive lipids with multiple activities, and further studies will be required to understand the interplay of the possible neuroprotective and anti-apoptotic mechanisms.

In summary, our work demonstrated that PUFA-plasmalogen LNP-based nanomedicines can control the kinetics of CREB phosphorylation/activation in view of recovery from neurotoxin-induced oxidative stress and neurodegeneration. The obtained new results on the regulation of CREB signaling pathway by PUFA-plasmalogen-loaded LNPs and lipid-peptide nanoassemblies revealed that the cellular response to nanoparticles is essentially dependent both on treatment time and type of the structural organization of nanoparticles (cubosome, hexosome, or vesicle). These features and the knowledge about the involved molecular mechanisms should contribute to more efficient design and fabrication of self-assembled nanomaterials for the treatment of neurological disorders, and potentially as alternatives strategies for the recovery from long COVID-19 neurological sequelae. Moreover, considering that CREB pathway comprises also a potential therapeutic approach against viral pathogenesis[66], the novel findings of this report may lead to future antiviral clinical applications.

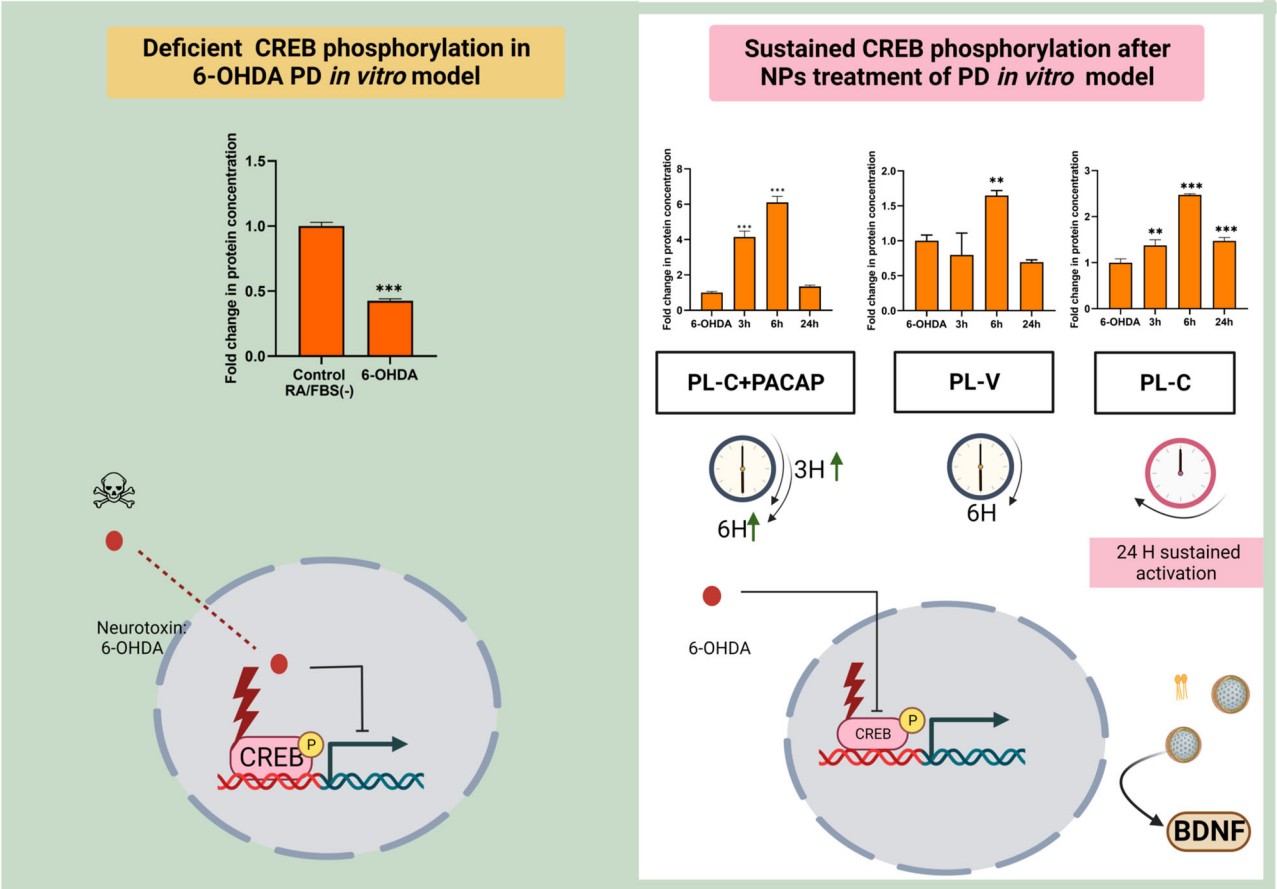

**Fig. 9 Summarized outcome of PUFA-plasmalogen-based LNPs and lipid-peptide (PUFA-plasmalogen-PACAP peptide) nanoassemblies-triggered CREB signaling in a Parkinson's disease (PD) model in vitro.** (Left panel) The levels of phosphorylated (activated) CREB are deficient in the diseased state (a PD model generated with 6-OHDA-induced oxidative stress in 24 h-starved differentiated SH-SY5Y cells) and cannot be fully recovered by vesicular LNP administration (middle histogram PL-V in the right panel). (Right panel) Nonlamellar liquid crystalline LNPs (PL-C) induce time-dependent signal processing pathways in the PD neuronal cell response as evidenced by sustained CREB phosphorylation (right histogram PL-C). The time response to treatment by lipid-peptide nanoassemblies (PL-C + PACAP) depends on cell-penetrating properties of the studied peptide and can be enhanced with regard to stimulation by PL-V.

## Materials and methods

**Materials**. The PUFA-plasmalogen (vinyl ether) derivative 1-(1Z-octadecenyl)-2-docosahexaenoyl-sn-glycero-3-phosphoethanolamine (PL-DHA-PE) and the fluorescent lipid 1,2-dioleoyl-sn-glycero-3-phosphoethanolamine-N-(lissamine rhodamine B sulfonyl) (ammonium salt) (18:1 Liss Rhod PE) were purchased from Avanti Polar Lipids, Inc. (Alabama). The composition of the scallop-derived plasmalogen extract was characterized by the provider as a mixture of ethanolamine vinyl ether phospholipid (49.4%), choline vinyl ether phospholipid (24.9%), cholesterol (16.0%), and ceramide aminoethyl phosphonate (9.7%). This natural plasmalogen combination with 70% vinyl ether phospholipid content is referred as scPL70. Monoolein (MO, C18:1c9, powder, ≥99%), retinoic acid (RA), vitamin E (VitE), 2,6-di-tert-butyl-4-methylphenol (BHT), and D-α-tocopherol polyethylene glycol-1000 succinate (TPGS-PEG$_{1000}$) were purchased from Sigma-Aldrich. Water of MilliQ quality (Millipore Corp., Molsheim, France) was used for preparation of a phosphate buffer solution (NaH$_2$PO$_4$/Na$_2$HPO$_4$, $1 \times 10^{-2}$ M, pH 7, p.a. grade, Merck).

**Lipid nanoparticle formulation**. Lipid nanoparticles (LNPs) were prepared by the method of hydration of a lyophilized thin lipid film followed by physical agitation in excess aqueous phase[46]. The composition of PL-V LNPs comprised plasmalogen scPL70, vitamin E, and TPGS-PEG$_{1000}$. The PL-H LNP formulation included MO, plasmalogen PL-DHA-PE, vitamin E, Q$_{10}$, and TPGS-PEG$_{1000}$, and the PL-C LNPs included MO, plasmalogen PL-DHA-PE, vitamin E, and TPGS-PEG$_{1000}$. Blank LNPs were prepared with MO and TPGS-PEG$_{1000}$. The protocol included a dry film formation after mixing lipid, stabilizer, and other components, rehydrating the film with an excess buffer solution, and using ultra-sonication to fragment the liquid crystalline assembly into particles. The buffer contained the antioxidant BHT, which ensured the oxidative stability of the aqueous formulations by suppressing the formation of lipid hydroperoxides. The phosphate/BHT buffer medium was purged by nitrogen gas (to eliminate the dissolved oxygen) and filtered through a 0.2 μm sterile filter (Millipore Corp.).

**Synchrotron small-angle X-ray scattering (SAXS)**. The dispersed nanoparticle samples were sealed in X-ray capillaries with diameters 1.5 mm. The methodology of the performed SAXS measurements at the SWING beamline of synchrotron SOLEIL (Saint Aubin, France) was analogous to the previously developed method by us[47]. The samples were placed in a designed holder with (X, Y, Z) positioning. Temperature was 22 °C. The chosen sample-to-detector distance accounted for the soft matter sample

properties. The X-ray beam spot size on the samples was $375 \times 25\,\mu m^2$. The patterns were recorded with a two-dimensional Eiger X 4 M detector (Dectris, Baden-Daettwil Switzerland) at 12 keV allowing measurements in the $q$-range from 0.0179 to 2.18 Å$^{-1}$. The $q$-vector was defined as $q = (4\pi/\lambda)$ sinθ, where $2\theta$ is the scattering angle. The synchrotron radiation wavelength was $\lambda = 1.033$ Å and the exposure time was 500 ms. The $q$-range calibration was done using a standard sample of silver behenate (repeat spacing $d = 58.38$ Å). An average of five spectra per sample was acquired. Data processing of the recorded 2D images was performed by the FOXTROT software as previously indicated[47].

**Cryogenic transmission electron microscopy (cryo-TEM)**. We previously established the methodology of the cryo-TEM imaging of liquid crystalline nanostructures[47]. In brief, a sample droplet of 2 μL was put on a lacey carbon film covered copper grid (Science Services, Munich, Germany), which was hydrophilized by glow discharge (Solarus, Gatan, Munich, Germany) for 30 s. Most of the liquid was then removed with blotting paper, leaving a thin film stretched over the lace holes. The specimen was instantly shock frozen by rapid immersion into liquid ethane and cooled to approximately 90 K by liquid nitrogen in a temperature and humidity controlled freezing unit (Leica EMGP, Wetzlar, Germany). The temperature and humidity were monitored and kept constant in the chamber during all sample preparation steps. The specimen was inserted into a cryo-transfer holder (CT3500, Gatan, Munich, Germany) and transferred to a Zeiss EM922 Omega energy-filtered TEM (EFTEM) instrument (Carl Zeiss Microscopy, Jena, Germany). Examinations were carried out at temperatures around 90 K. The TEM instrument was operated at an acceleration voltage of 200 kV. Zero-loss-filtered images (DE = 0 eV) were taken under reduced dose conditions (100–1000 e/nm$^2$). The images were recorded digitally by a bottom-mounted charge-coupled device (CCD) camera system (Ultra Scan 1000, Gatan, Munich, Germany) and combined and processed with a digital imaging processing system (Digital Micrograph GMS 1.9, Gatan, Munich, Germany). The sizes of the investigated nanoparticles were in the range or below the film thickness and no deformations were observed. The images were taken very close to focus or slightly under the focus (some nanometers) due to the contrast enhancing capabilities of the in-column filter of the employed Zeiss EM922 Omega. In EFTEMs, the deep underfocused images can be totally avoided.

**Cell culture**. The human neuroblastoma SH-SY5Y cells were grown in Dulbecco's Modified Eagles Medium (DMEM Sigma, France) completed with 1% penicillin-streptomycin (Sigma) and 10% (v/v) Fetal Bovine Serum (FBS, Sigma). The cell culture was maintained in a humidified 5% $CO_2$ atmosphere at 37 °C. The SH-SY5Y cells were subcultured every 3 days. Briefly, cells were incubated with 3 mL trypsin (0.05% w/v)/EDTA (0.02% w/v) for 5 min to detach the adhesive cells following washing with a sterile Phosphate-Buffered Saline (PBS) deficient of calcium ions. Fresh complete medium DMEM (7 mL) was added to stop the trypsin reaction. The pellet was collected by ultracentrifugation at 200 rpm 20 °C for 5 min and diluted with 10 mL fresh complete medium. For subsequent experiments, the SH-SY5Y cells were differentiated to a neuronal phenotype using 10 μM RA for 5 days. The differentiation was initiated at about 70–80% cell confluence, on the second day of seeding, by replacing the culture medium by 15 mL fresh complete medium containing 10 μM RA. The cells investigated in this work were collected from passage numbers less than 25.

**Human tyrosine hydroxylase (TH) ELISA assay**. The human TH levels in supernatants of treated cells were measured by the commercial ELISA kit Human TH ELISA kit (Cat. #KTE60519, Abbkine). The ELISA assay was performed according to the manufacturer's instructions. In short, the supernatants were diluted with 40 μL dilution buffer and added to pre-coated plates for 45 min incubation at 37 °C. After that, the plate was washed three times with the washing solution and incubated with HRP-conjugated detection antibody for 30 min at 37 °C. Then, the plate was washed and incubated with mixed chromogen A and B reagents for 15 min at 37 °C. Finally, the color was developed with a stop solution. The absorbance was then measured spectrophotometrically at 450 nm to quantify the levels of TH.

**Cell viability assay**. SH-SY5Y cells (10,000 cells/well) were seeded in 96-well plates incubated overnight at 37 °C 5% $CO_2$ for adhesion. Then, the medium was replaced with a complete medium containing 10 μM RA. The cells were then incubated for 5 days, during which they were monitored regularly for signs of differentiation. After 5 days, the RA-differentiated SH-SY5Y cells were deprived of FBS during 24 h and treated with the desired formulations. RA-differentiated SH-SY5Y cells deprived of FBS during 24 h were used as controls referred to as RA/FBS(-). After treatment, the medium was removed, and then 20 μL MTT (3-(4,5-dimethylthiazol-2-yl)-2,5-diphenyltetrazolium bromide) reagent (5 mg/mL) was added to each well and incubated for an additional 1 h. After that, the medium was removed, and the formed formazan crystals were dissolved in DMSO. The absorbance was then determined spectrophotometrically at 570 nm (LT-5000 MS, Labtech) to quantify the cell viability. The MTT solution with 5 mg/mL concentration was prepared in PBS and was filtered before use.

**Mitochondrial membrane potential assay**. Mitochondrial membrane potential was probed by a JC-10 dye assay (Abcam, CA, USA). The SH-SY5Y cells were cultured in 6-well plates at the density of $3 \times 10^5$ cells/well. After 5 days of differentiation by 10 μM retinoic acid, the cells were treated with 10 μM lipid nanoparticle formulations for 24 h in the lack of serum. Incubation with 5 μM trifluoromethoxy carbonylcyanide phenylhydrazone (FCCP) for 30 min was studied as a positive control. RA-differentiated cells RA/FBS(-) were used as a negative control. The cells were stained by JC-10 dye for 15 min and protected from light. Washing with PBS (2 times) and 0.5 mL trypsin-EDTA solution was done in each well for detaching the cells. The cell suspension was transferred into an Eppendorf tube and ultracentrifuged at 200 rpm for 5 min. The pellet was separated from the supernatant and washed 2 times with PBS. The samples were measured using an Accuri C6 flow cytometer (Becton Dickinson). The green fluorescence was recorded at the FL1 channel, whereas the red fluorescence was recorded at the FL2 channel. The experimental data were expressed as fluorescence intensity ratios [red (FL2)/green (FL1)]. The employed lipophilic cationic carbocyanine dye accumulates in mitochondria and forms J-aggregates characterized by red fluorescence (FL2). Mitochondrial membrane depolarization (provoked by neurotoxicity or mitochondrial apoptosis) causes leakage of the monomeric green dye form from the mitochondria to the cytoplasm, producing green fluorescence signal (FL1).

**Flow cytometry measurement of reactive oxygen species (ROS)**. Changes in the ROS levels were measured by flow cytometry as previously described[67]. The DCFHDA dye probe was used for the detection of intracellular ROS in cells subjected to 6-OHDA exposure. The excitation was performed at a wavelength of

488 nm. The fluorescence emission of the dye probe was collected in the FL1 533 nm region. DCFH-DA is a nonpolar permeable dye, which is converted to DCFH by cellular esterases into a nonfluorescent polar derivative. DCFH is then converted to highly fluorescent DCF when oxidized by intracellular ROS and other peroxides. The DCF accumulation was measured by the increase in the fluorescence at 530 nm when the sample is excited at 488 nm. The measured intensity is proportional to the concentration of ROS inside the cells. In brief, the cells were seeded in a 6-well plate at a density of $5 \times 10^6$ cells. On the second day, the complete medium was replaced with a 10 μM RA-containing complete medium. The RA-containing complete medium was changed every three days. On the sixth day, the old medium was removed, and the cells were treated with RA-containing FBS-free medium RA/FBS(-) containing different concentrations of 6-OHDA for 30 min. The cells were washed with PBS, harvested using trypsin, and centrifuged at 200 g for 5 min at 4°C. The cells were then resuspended in PBS, followed by another centrifugation step at 200 g for 5 min at 4°C. After discarding the supernatant, the cells were incubated with 10 μM DCFH-DA in DMEM without FBS at 37 °C for 15 min. The cells were washed twice and subsequently measured using flow cytometry (BD Accuri C6).

**Cellular uptake of nanoparticulate assemblies**. The SH-SY5Y cells were cultured in 6-well plates at a density of $3 \times 105$ cells/well. After 5 days of differentiation by 10 μM, the medium was replaced with a fresh medium RA/FBS(-) containing fluorescently-labeled nanoassemblies for different times. After the incubation period, the cells were washed three times with cold PBS, detached by an appropriate buffer, and centrifuged at 200 g for 5 min. The pellet was separated from the supernatant and diluted in 1 mL PBS. The fluorescence of the samples was recorded in the FL2 channel of the flow cytometer (BD Accuri C6).

**Quantification of phosphorylated CREB, AKT, and ERK proteins by enzyme-linked immunosorbent assays (DuoSet IC ELISA)**. Human phosphorylated CREB (p-CREB) was quantified in cell lysates of nanoparticle-treated cells using a Phospho-CREB (S133) DuoSet IC ELISA kit (Cat. #. DYC2510-5, Bio-Techne Ltd./R&D Systems, UK), which is specific for CREB phosphorylation at S133. Phospho-ERK1 (T202/Y204)/ERK2 (T185/Y187) DuoSet IC ELISA (Cat. #. DYC1018B-5, Bio-Techne Ltd./R&D Systems, UK), and Phospho-AKT (S473) Pan Specific DuoSet IC ELISA kits (Cat. #. DYC887B-5, Bio-Techne Ltd./R&D Systems, UK) were used for quantification of p-ERK1/2 and p-AKT in human cell lysates, respectively. The cell lysates were prepared with a specific lysis buffer and the analysis was performed with the use of appropriate DuoSet ELISA ancillary reagent kits (Bio-Techne Ltd./R&D Systems, UK).

The concentrations of phosphorylated CREB at Ser-133, phosphorylated AKT, and phosphorylated ERK were determined following the manufacturer's instructions. Briefly, the 96-well plates were pre-coated with a monoclonal antibody specific for p-CREB, p-AKT or ERK1/2. The protein standards and cell lysate samples were pipetted and incubated into the wells for 2 h and any free protein (p-CREB, p-AKT, or ERK1/2) present was bound by the immobilized antibody. An enzyme-linked monoclonal antibody which is specific for the measured protein, was added to the wells for 1 h. All the incubation stages were conducted at room temperature. Then, a procedure of a total of 3 washes with washing buffers was performed to remove any unbound antibody-enzyme reagent. A substrate solution was added to the wells for 30 min. The color was developed in the wells proportionally to the amount of protein in the initial step. The plates were washed and developed with a substrate/chromogen

solution. The color development was discontinued with a stop solution and the color intensity was measured by a plate reader. The absorbance was measured spectrophotometrically at 450 nm within 30 min after stopping the reaction to quantify the levels of phosphorylated CREB, AKT, and ERK proteins.

**Total protein concentration**. The total protein concentration in the cell lysates was measured by Bradford Protein Assay using 96-well plates. A stock solution of BSA (bovine serum albumin) with a concentration 2 mg/mL in PBS was used, from which a calibration range of seven BSA concentrations was prepared. 20 μL of BSA standard solutions or cell lysate samples were mixed with 180 μl Bradford reagent in the wells (Sigma, B6916-500ML). The absorbance of each well was measured using a microplate reader at a wavelength of 595 nm. The optical density of a black well was subtracted. The measured total protein content was used for normalization of the protein amounts determined by quantitative ELISA in the studied samples.

**Statistical analysis**. Data were expressed as the mean ± SD of at least three biological replicates. The Dunnett test was used for multiple comparisons. The differences between two groups were assessed using Student's t-test for normally distributed data. $P < 0.05$ indicated significance. In ELISA, one-way ANOVA was performed to detect significant differences between groups. Additionally, the Sigmoidal 4PL model was used to explore relationships between cells and protein expression. All statistical analysis was performed using GraphPad Prism 9.

**Reporting summary**. Further information on research design is available in the Nature Portfolio Reporting Summary linked to this article.

## Data availability
The source data used for generating the graphs is provided in "Supplementary Data 1" and the data supporting the study's findings are accessible upon reasonable request from the corresponding authors.

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

## Acknowledgements

The performed research was funded by the projects "Structural Dynamics of Biomolecular Systems" (ELIBIO) (CZ.02.1.01/0.0/0.0/15_003/0000447) and "Advanced research using high-intensity laser produced photons and particles" (CZ.02.1.01/0.0/0.0/16_019/0000789) from the European Regional Development Fund. A.A. and B.A. acknowledge the scientific support by Dr. Thomas Bizien at the SWING beamline of Synchrotron SOLEIL (Saint Aubin, France) and the allocation of beam time through the projects 20210580 and 20201321. Y.W. acknowledges a PhD fellowship from MESRI (France). Y.D. was supported by a grant from the Wenzhou Institute, University of Chinese Academy of Sciences (Grant No. WIUCASQD2019005). M.D. was supported by the collaborative research center SFB840 of the German Science Foundation DFG (Germany). A.A. acknowledges membership in CNRS GDR2088 BIOMIM research network. The medical art images in Figs. 2 and 9 were generated with BioRender (*BioRender.com*).

## Author contributions

A.A. conceived the project and designed the research. Y.W., A.A., B.A., and M.D. performed the experiments and analyzed the data. B.A., Y.D., T.F., and M.S.H. provided resources and engineered custom-purified vinyl ether lipids. Y.W., B.A., and A.A. designed the figures. Y.W. prepared an initial draft. A.A. and Y.W. wrote the paper. All authors contributed to the discussion of the results included in the article.

## Competing interests

The authors declare no competing interests.
