## [Peer Review File · Communications Chemistry]

Reviewers' comments:

Reviewer #1 (Remarks to the Author):

The manuscript entitled "Sustained CREB Phosphorylation by Lipid-Peptide Liquid Crystalline nanoassemblies" by Wu and co-workers is of great importance. Different liquid crystalline structural organizations containing PUFA-Plasmalogens or PUFA-Plasmalogens and a peptide have been compared. It was demonstrated that they can mediate a beneficial effect on a cell culture model related to Parkinson's disease.

My comments are as follow:

1. The manuscript would benefit if some basic terms would be better explained for a more general readership: e.g.: What determines the Bragg peak maxima and what is the importance? Should one see the described maxima positions of the peaks in the plot, e.g. Page 9, peaks described in line 213 and 214 in the Fig 3b dark blue plot?
2. For the in vitro treatment part it appears of importance that the plasmalogen treatments are indeed in relation to the 5-OHDA treatment and not solely a treatment of the starvation. This has to be discussed and clarified and the controls should always be also 24 hours starved. The figure 5 A and 6 A must show SHSY5Y cells after RA differentiation for 5 days and then 24 hours starvation.
3. What is actually the molar amounts of PL-DHA-PE and monoolein in the stable nonlamellare liquid crystalline phase?
4. Page 15 line 346: "...slightly increases this .." even if it is not significant the exact p-value can be presented and it can be stated in accordance to the p value either as it is or end the sentence with "but do not reach statistical significance (P=0,xx).
5. Related to point 2; Fig 5 and Fig6: It is indicated that all the experiments were conducted after cellular differentiation for 5 days followed by 24 hour of incubation in a serum-free medium. It should be clarified if indeed the control was always also for 24 hours in lipid deficient medium. Why is the lipid depleted medium treatment after the application of 6-OHDA? Was the pre-treatment during the differentiation?
6. Fig 6b: The figure legend and the figure do not fit. If indeed a treatment is intended after the toxic incubation for more than 30 Min this will require other controls.
7. The manuscript would benefit from a section of the limitations of this study do avoid the expression of an over interpretation of the cell culture model.

Reviewer #2 (Remarks to the Author):

This study deals with the therapeutic application of the nanomolecules consisting of plasmalogen and PACAP for PD model by examining the change in the intracellular signaling pathway, such as CREB, AKT, and ERK. This paper includes very interesting information about the therapeutics for neurological and developmental disorders though I have some concerns. Please address the following questions.

Major points:

Results

1. Line 347: The average of the fluorescence intensity ratio of a dye probe is slightly higher in 6-OHDA treated cells, but there is no significant difference between the control and 6-OHDA groups. I think this sentence should be deleted if there is no tendency to be higher in the fluorescence intensity ratio in the 6-OHDA group compared to the control statistically.
2. Figure 5g: please show the significance level regarding the fold change in the CREB phosphorylation.
3. Cellular viability is higher in the PL-H group than in the control. Does PL-H also prevent apoptosis of the RA-treated cells in the serum free medium? If so, PL-C may inhibit 6-OHDA induced apoptosis whereas PL-H might inhibit 6-OHDA induced apoptosis but also general apoptosis. Please discuss it.
4. According to the Fig. 7, only two samples were examined in the experiment of the kinetic changes of phosphorylation of CREB, AKT, and ERK. For the comparison between groups statistically, three or more samples are necessary. Please verify the number of the samples.
5. Phosphorylation of AKT was promoted by administration of PL-V, and PACAP-LNP nano-assemblies also increased pAKT level at 6hours after administration. On the other hand, the author mentioned that PL had protective effects on the PD model by activation of CREB mediated through mainly ERK pathway but not AKT pathway because PL-C and PL-H did not increase pAKT. This explanation may make the readers confused. Please discuss further about the signaling pathway to activate CREB.

Discussion

6. The graph of the kinetic change CREB in figure 9 may be different from that in figure 8. Peak of phosphorylation is at 6 hours after PACAP-LNP administration in figure 8. Hence, I think, PACAP does not make CREB activation faster. Please discuss further about it.

Minor points:

Methods

1. Please specify the multiple comparison test used in the ANOVA.

Reviewer #3 (Remarks to the Author):

The manuscript by Wu et al. reports on the impact of designed dual-loaded LNPs and bioactive lipid-peptide nanoassemblies, with different liquid crystalline structural organizations, on the kinetics of CREB activation and recovery from oxidative stress in a Parkinson's disease (PD) model in vitro.

The research presented in this manuscript has been well conducted with very apt testing and scientifically sound results and discussion. As such I have minor comments.

The authors are invited to comment on the stability of the lipid nanoparticles. For instance, membrane

integrity studies using a hydrophilic marker (e.g. calcein) would benefit the manuscript.

Microcalorimetry studies would offer important information on the miscibility of the actives with lipids and would further augment the results obtained from SAXS and Cryo-TEM studies

SAXS studies were performed at 25oC and not at 37oC which is closer to the in vivo conditions.

Moreover, time resolved studies would benefit the manuscript shedding light to the behavior of the formulation over time. Please comment on this.

Reply to Reviewers' comments

Manuscript ID: **COMMSCHEM-23-0328-T**

Title: *Sustained CREB Phosphorylation by Lipid-Peptide Liquid Crystalline Nanoassemblies*

We thank the Reviewers for the insightful comments and suggestions, which helped to considerably improve our manuscript. The changes made are marked in yellow in the revised manuscript. The performed revisions included some complementary experiments as described below.

Herewith are the point-by-point answers to the Reviewers' comments.

Reviewer #1

The manuscript entitled “Sustained CREB Phosphorylation by Lipid-Peptide Liquid Crystalline nanoassemblies” by Wu and co-workers is of great importance. Different liquid crystalline structural organizations containing PUFA-Plasmalogens or PUFA-Plasmaligens and a peptide have been compared. It was demonstrated that they can mediate a beneficial effect on a cell culture model related to Parkinson's disease. My comments are as follow:

1. The manuscript would benefit if some basic terms would be better explained for a more general readership: e.g.: What determines the Bragg peak maxima and what is the importance? Should one see the described maxima positions of the peaks in the plot, e.g. Page 9, peaks described in line 213 and 214 in the Fig 3b dark blue plot?

Answer: Thank you for your positive appreciation of our paper and your questions.

(I) We added the following text in the revised version:

“The Bragg peak position maxima in a SAXS pattern are determined by the periodicity of the structures that scatter the X-rays. Their q -vector positions are used for the determination of the structural parameters and the lattice spacing of the generated three-dimensional (3D) liquid crystalline organizations. The absence of Bragg diffraction peaks in the recorded scattering plots implies that no periodic 3D assemblies are formed upon hydration and dispersion of the lipid phases into nanoparticles.”

(II) Following your suggestion, Figures 3b and 3c were modified by adding arrows that point out and visualize the positions of the Bragg peaks, which are discussed in the text.

We also added a sentence in the Figure 3 caption: “The arrows indicate the positions of the Bragg peaks discussed in the text.”

(III) Regarding the importance of the structural features, determined from the positions of the resolved Bragg diffraction peaks, we explained in the revised text that:

“The established nanochannel properties suggest the noteworthy capacity of this non-lamellar type of LNPs for incorporation and encapsulation of guest molecules, e.g. peptides of therapeutic significance such as PACAP.”

“The smaller a_{HII} lattice parameter as compared to the cubic a_0 lattice parameter implies that the density of the lipid building blocks (lipid tubes or bicontinuous bilayer membranes) is higher in the supramolecular structure of the hexosome PL-V nanocarriers as compared to the cubosome PL-C ones. This structural feature may influence the compound release and the interaction of the two types of LNPs with the biological membrane interfaces”.

2. For the *in vitro* treatment part it appears of importance that the plasmalogen treatments are indeed in relation to the 5-OHDA treatment and not solely a treatment of the starvation. This has to be discussed and clarified and the controls should always be also 24 hours starved. The figure 5 A and 6 A must show SHSY5Y cells after RA differentiation for 5 days and then 24 hours starvation.

Answer: Thank you for your comments and suggestions.

(I) The images of RA-differentiated cells in Figures 5A and 6A depict cells that were subjected to 5 days of RA-differentiation followed by 24h of starvation before the addition of neurotoxin 6-OHDA for 30 min. We re-labeled the morphological images in Figure 5 and Figure 6 as "RA/FBS(-)" for better clarity and understanding.

We use this treatment because it produces environmentally sensitive differentiated SH-SY5Y cells with augmented responsiveness to 6-OHDA. Indeed, the 24-hour period of starvation for SH-SY5Y cells does not induce significant apoptosis but rather makes the cells resemble a neuronal cell phenotype. This can be observed through the morphological changes as shown in Figure S1.

“An *in vitro* PD model was created by neurotoxin [6-hydroxydopamine (6-OHDA)]- induced oxidative stress using 24h-starved differentiated human neuroblastoma SH-SY5Y cells of a neuronal phenotype.”

(II) We included optical images of RA-differentiated cells taken after 5 days, both with and without starvation (Figure S1).

Figure S1. Optical micrographs of SH-SY5Y cellular morphologies showing the effect of starvation before and after treatment by retinoic acid (RA 10µM) for differentiation into a neuronal phenotype. The cell culture conditions correspond to

- non-differentiated cells (no RA) grown in serum-containing medium (FBS+);
- non-differentiated cells (no RA) exposed for 24h to FBS-free medium (FBS-), *i.e.* 24h starvation of non-differentiated cells;
- RA-differentiated cells for 5 days in FBS-complete medium (RA/(FBS(+))); and
- RA-differentiated cells (5 days) that are further exposed for 24h to RA-containing FBS-free medium (RA/FBS (-)), *i.e.* 24h starvation in RA-containing medium.

(III) We included a new figure in the Supplementary Information to show the effect on CREB phosphorylation of non-differentiated, RA-differentiated, and starved RA-differentiated SH-SY5Y cells (see Figure S2 below).

Figure S2. Effect of SH-SY5Y cellular differentiation and starvation on the fold changes in phosphorylated CREB (pCREB) levels before the oxidative damage by 6-OHDA. The cell culture conditions correspond to:

- non-differentiated SH-SY5Y cells grown in a medium containing FBS, *i.e.* no RA/FBS(+);
- non-differentiated SH-SY5Y cells exposed for 24h to FBS-free medium, *i.e.* no RA/FBS(-);
- RA-differentiated SH-SY5Y cells grown in FBS-containing complete medium for 5 days, *i.e.* RA/FBS(+);
- RA-differentiated SH-SY5Y cells exposed for 24h to FBS-free medium, *i.e.* RA/FBS(-).

Phosphorylated CREB was quantified by ELISA. Statistical significance is indicated as follows: * $P \leq 0.05$, ** $P \leq 0.01$, and *** $P \leq 0.001$.

(IV) In the revised manuscript, we have provided an explanation for why all the experiments were conducted after a 24-hour period of starvation.

“The deprivation of serum (24h) from the culture medium of the RA-differentiated SH-SY5Y cells was done based on published findings that it increases the cellular susceptibility to the environment and the applied treatment^{1,2}. Serum starvation facilitates the induction of neurite formation and prevents excessive growth of S-type SH-SY5Y cells. Moreover, SH-SY5Y cells when subjected to nutrient deprivation, display a higher responsiveness to 6-OHDA that induces apoptosis and cell death. The morphologies of differentiated SH-SY5Y cells, both before and after 24h starvation, are shown in Figure S1 of the *Supplementary Information*.”

(V) The following text was included in the caption of Figure 2 in order to precise how the neurodegeneration model was *in vitro* generated in this work:

“In the PD model created *in vitro* (through cellular SH-SY5Y differentiation by retinoic acid (RA) for 5 days, starvation for 24h in RA-only medium, and incubation with the neurotoxin 6-OHDA for 30min).”

(VI) The labels in the Figures presenting *in vitro* results about the cellular PD model (Fig. 5 and Fig. 6) were modified in order to clarify that the control sample RA/FBS(-) comprises 24h-starved RA-differentiated SH-SY5Y cells.

Fig. 5. *In vitro* PD model induced by 6-OHDA with differentiated SH-SY5Y cells following 24h-starvation in RA-only medium.

Fig. 6. Regenerative effects of PUFA-plasmalogen-based particles on the *in vitro* PD model.

3. What is actually the molar amounts of PL-DHA-PE and monoolein in the stable nonlamellar liquid crystalline phase?

Answer: Thank you for your question.

The molar amount of PL-DHA-PE in the self-assembled amphiphilic mixture is determined from the mass quantities indicated in Table S1 of the Supplementary Information.

We added the following text in the revised version:

“The molar amount of PL-DHA-PE with regard to monoolein was 15 mol.%”

4. Page 15 line 346: “..slightly increases this ..” even if it is not significant the exact p-value can be presented and it can be stated in accordance to the p value either as it is or end the sentence with “but do not reach statistical significance (P=0,xx).

Answer: Thank you for your comment. The text was modified as follows.

”Figure 5d demonstrates that the mitochondrial membrane depolarization by 6-OHDA slightly increases compared to the control group, but does not reach statistical significance (P=0,13).”

5. Related to point 2; Fig 5 and Fig6: It is indicated that all the experiments were conducted after cellular differentiation for 5 days followed by 24 hour of incubation in a serum-free medium. It should be clarified if indeed the control was always also for 24 hours in lipid deficient medium. Why is the lipid depleted medium treatment after the application of 6-OHDA? Was the pre-treatment during the differentiation?

Answer: Thank you for your insightful comment.

We confirm that all results were obtained with a control sample of RA/FBS(-) (serum deficient medium). The new Figure S3 in the Supplementary information shows the comparison of the pCREB, pAKT, and pERK fold changes due to the exposure of differentiated 24h-starved SH-SY5Y cells to the neurotoxin (6-OHDA) in serum-free medium.

Figure S3. Fold changes in the phosphorylation levels of CREB, AKT, and ERK proteins following 6-OHDA (30 min) treatment of 24h-starved differentiated SH-SY5Y cells, quantified by ELISA assays. The control RA/FBS(-) corresponds to RA-differentiated SH-SY5Y cells exposed for 24h to FBS-free medium. Statistical significance is as follows: * $P \leq 0.05$, ** $P \leq 0.01$, and *** $P \leq 0.001$.

We included a new Figure S2 in the Supplementary information to show the effect of differentiation and starvation of the SH-SY5Y cells on the pCREB levels before the addition of the neurotoxin (6-OHDA).

Figure S2. Effect of SH-SY5Y cellular differentiation and starvation on the fold changes in phosphorylated CREB (pCREB) levels before the oxidative damage by 6-OHDA. The cell culture conditions correspond to:

- non-differentiated SH-SY5Y cells grown in a medium containing FBS, i.e. no RA/FBS(+);
- non-differentiated SH-SY5Y cells exposed for 24h to FBS-free medium, i.e. no RA/FBS(-);
- RA-differentiated SH-SY5Y cells grown in FBS-containing complete medium for 5 days, i.e. RA/FBS(+);
- RA-differentiated SH-SY5Y cells exposed for 24h to FBS-free medium, i.e. RA/FBS(-).

Phosphorylated CREB was quantified by ELISA. Statistical significance is indicated as follows: * $P \leq 0.05$, ** $P \leq 0.01$, and *** $P \leq 0.001$.

In our methodology, the cells were treated with 10 μM lipid nanoparticle formulations in the lack of serum. When studying lipid nanoparticles (LNPs), we use lipid-deficient medium RA/FBS(-) in order to evaluate the specific effect of the LNP treatment and to avoid the overlapping global effects of other compounds.

The 24h starvation pre-treatment enhances the neuronal-like characteristics of the differentiated SH-SY5Y cells, thus making them more sensitive to the tested formulations.

6. Fig 6b: The figure legend and the figure do not fit. If indeed a treatment is intended after the toxic incubation for more than 30 Min this will require other controls.

Answer: Thank you for your comment. The text has been modified.

7. The manuscript would benefit from a section of the limitations of this study do avoid the expression of an over interpretation of the cell culture model.

Answer: Thank you for your comment and suggestion.

Additional text was incorporated into the Discussion to point out the limitations.

The limitations of this study are associated with the selected in vitro disease model of neurodegeneration. In the absence of previously published information concerning the spatio-temporal kinetics of CREB activation by lipid-peptide nanoassemblies and considering that CREB phosphorylation typically lasts about 20 minutes for signaling pathway activation by free drugs, we developed our in vitro neurodegeneration model through short-term exposure to the neurotoxin 6-OHDA. To enhance the responsiveness of the RA-differentiated SH-SY5Y cells to short-term damage induced by 6-OHDA, we exposed the cells to 24 h of pre-treatment in serum-depleted medium, specifically, 24 h of starvation in an RA-only medium. The obtained data about LNP-mediated neuronal cell recovery from oxidative stress indicate that cellular viability is higher in the PL-H group than in the control RA/FBS(-) sample, as cellular viability is also higher in the PL-H group than in the 6-OHDA group. Thus, PL-H may be able to inhibit 6-OHDA-induced apoptosis as well as general apoptosis, likely because LNPs can serve as a

nutrient source for the cells. We have previously reported that PUFA-based LNPs (e.g., cubosomes loaded with DHA) promote the cellular survival of 24h-starved RA-differentiated SH-SY5Y cells and increase their viability⁶⁴. The present work confirms that plasmalogen-based LNPs may favour also recovery from short-term oxidative damage caused by neurotoxin 6-OHDA in addition to recovery from starvation that leads to cellular apoptosis.

The exact signaling pathway and dynamics of CREB activation by plasmalogen species in damaged neuronal cells is still not well understood. PUFA-plasmalogens have free-radical scavenging and antioxidant properties. In addition, they have been suggested to activate G-protein coupled receptors in the cellular lipid membranes (Fig. 2), thus acting as hormones⁶⁵. Therefore, both membrane receptor activation and intracellular neurotrophic signaling may be evoked to explain the effect of PL-based nanoassemblies on CREB phosphorylation. Whereas the present study focused on the capacity of PUFA-plasmalogen-based nanoformulations to prolong the kinetics of CREB phosphorylation beyond 1h, PUFA-plasmalogens are bioactive lipids with multiple activities, and further studies will be required to understand the interplay of the possible neuroprotective and anti-apoptotic mechanisms.

Reviewer #2

This study deals with the therapeutic application of the nanomolecules consisting of plasmalogen and PACAP for PD model by examining the change in the intracellular signaling pathway, such as CREB, AKT, and ERK. This paper includes very interesting information about the therapeutics for neurological and developmental disorders though I have some concerns. Please address the following questions.

Major points:

Results

1. Line 347: The average of the fluorescence intensity ratio of a dye probe is slightly higher in 6-OHDA treated cells, but there is no significant difference between the control and 6-OHDA groups. I think this sentence should be deleted if there is no tendency to be higher in the fluorescence intensity ratio in the 6-OHDA group compared to the control statistically.

Answer: Thank you for your comment.

The text has been modified, and an exact P-value was added to emphasize that there is no significant difference between the control and the 6-OHDA groups.

2. Figure 5g: please show the significance level regarding the fold change in the CREB phosphorylation.

Answer: Thank you for your comment.

The significance level has been added in figure 5 g.

3. Cellular viability is higher in the PL-H group than in the control. Does PL-H also prevent apoptosis of the RA-treated cells in the serum free medium? If so, PL-C may inhibit 6-OHDA induced apoptosis whereas PL-H might inhibit 6-OHDA induced apoptosis but also general apoptosis. Please discuss it.

Answer: Thank you for your comment.

One possible explanation of the viability exceeding 100% could be that PL-H prevents apoptosis of the RA-treated cells in the serum-free medium.

Encinas *et al.* have demonstrated that SH-SY5Y cells exhibit signs of apoptosis 6 and 24 hours after serum removal, as evidenced by caspase activity and TUNEL assay⁵. Literature reports have shown that plasmalogens can significantly inhibit apoptosis in SH-SY5Y cells⁶.

On the other side, Thomson *et al.* have indicated that 24-h FBS starvation enhances the neuronal characteristics of SH-SY5Y cells, but they did not find effects of serum deprivation on the expression of genes linked to apoptosis¹.

Another explanation is that the viability of PL-H group exceeds 100% because PL-H LNPs serve as a nutrient source, which stimulates cell growth. Although we differentiated SH-SY5Y cells with 5 days of RA treatment, they may still retain some proliferation capacity, given their neuroblastoma cell origin.

4. According to the Fig. 7, only two samples were examined in the experiment of the kinetic changes of phosphorylation of CREB, AKT, and ERK. For the comparison between groups statistically, three or more samples are necessary. Please verify the number of the samples.

Answer: Thank you for your comment.

Figure 7 was revised. For clarity, the panels about the response of the normal differentiated cells (not treated by 6-OHDA) were extracted and moved to the Supplementary Information. The presented additional data (with statistical validity) compare the outcome of the treatment of the *in vitro* PD model by different types of LNPs, for which the 6-OHDA damage represents the initial state.

6-OHDA induced PD model with SH-SY5Y cells

Fig. 7. Kinetic changes of phosphorylation of CREB, AKT, and ERK proteins following treatment with LNPs of the *in vitro* PD model.

We included a new Figure S3 in the Supplementary information to show the effect of the 24h-starvation condition on the phosphorylation levels of CREB, AKT, and ERK proteins following 30 min incubation with 6-OHDA (200 μ M).

Figure S3. Fold changes in the phosphorylation levels of CREB, AKT, and ERK proteins following 6-OHDA (30 min) treatment of 24h-starved differentiated SH-SY5Y cells, quantified by ELISA assays. The control RA/FBS(-) corresponds to RA-differentiated SH-SY5Y cells exposed for 24h to FBS-free medium. Statistical significance is as follows: * $P \leq 0.05$, ** $P \leq 0.01$, and *** $P \leq 0.001$.

The amelioration of the CREB phosphorylation levels of 24h-starved RA/FBS(-) control group after 24h-treatment by different LNPs (PL-V, PL-C and PL-H) is demonstrated in Figure S4. In all cases, LNPs increase the pCREB levels in comparison to the 6-OHDA-induced group.

Figure S4. Fold changes in the phosphorylation levels of CREB protein (pCREB) after exposure to 6-OHDA (30 min, 200 μ M) or LNPs (24h, 10 μ M) incubation. Nanoformulations of PL-V, PL-H, and PL-C are compared. Protein phosphorylation levels were quantified using an ELSA assay with regard to an RA/FBS(-) control sample (i.e. differentiated SH-SY5Y cells deprived

of serum for 24h). Statistical significance is as follows: * $P \leq 0.05$, ** $P \leq 0.01$, and *** $P \leq 0.001$.

5. Phosphorylation of AKT was promoted by administration of PL-V, and PACAP-LNP nano-assemblies also increased pAKT level at 6hours after administration. On the other hand, the author mentioned that PL had protective effects on the PD model by activation of CREB mediated through mainly ERK pathway but not AKT pathway because PL-C and PL-H did not increase pAKT. This explanation may make the readers confused. Please discuss further about the signaling pathway to activate CREB.

Answer: Thank you for your comment.

We deleted from the manuscript all the text related to the confusing statement “The observed time dependence is better expressed for ERK signaling pathway.”

Discussion

6. The graph of the kinetic change CREB in figure 9 may be different from that in figure 8. Peak of phosphorylation is at 6 hours after PACAP-LNP administration in figure 8.

Answer: Thank you for your comment.

The confusion has arisen because of the different control samples used for the presentation of the Fold change in the protein expression levels.

In the revised version, the changes in the phosphorylation levels of CREB, AKT and ERK are determined by DuoSet IC ELISA with regard to the diseased state (6-OHDA), whereas the RA/FBS(-) control sample was used as a reference in some of the initially submitted figure panels. So, Figure 8 was modified as shown below.

Fig. 8. Internalization kinetics (top row) and kinetic changes of CREB, AKT, and ERK phosphorylation in an in vitro PD model treated by nanoassemblies of PL-C and PACAP (bottom row).

To closely observe the fold changes in CREB phosphorylation, we extracted the panels for the responses to PL-V (Fig.7a), PL-C (Fig. 7g), and PL-C+PACAP (Fig. 8d) treatment and organize them together for a comparison herewith:

At the 3h point, we observe an essentially enhanced absolute value of the Fold change in the expression of pCREB in the (PL-C+PACAP) group (value nearly equal to 4, right panel) as compared to the PL-C group (value around 1.4, middle panel) and the PL-V group (left panel with a much smaller value).

Hence, I think, PACAP does not make CREB activation faster. Please discuss further about it.

Thanks to your comment, we modified the term “faster” to “enhanced” when describing the effect of PACAP.

The revised Figure 9 shows now the following tendency after correcting the control sample in Figure 8 and the re-estimation of the data for the pCREB fold changes:

Fig. 9. Summarized outcome of PUFA-plasmalogen-based LNPs and lipid-peptide (PUFA-plasmalogen-PACAP peptide) nanoassemblies ...

Minor points:

Methods

1. Please specify the multiple comparison test used in the ANOVA.

Answer: Thank you for your comment.

We added the following text in the revised paper:

“The Dunnett test was used for multiple comparisons, and the Student t-test was employed for comparing two groups, both using Prism software”.

Reviewer #3

The manuscript by Wu et al. reports on the impact of designed dual-loaded LNPs and bioactive lipid-peptide nanoassemblies, with different liquid crystalline structural organizations, on the kinetics of CREB activation and recovery from oxidative stress in a Parkinson's disease (PD) model *in vitro*.

The research presented in this manuscript has been well conducted with very apt testing and scientifically sound results and discussion. As such I have minor comments.

The authors are invited to comment on the stability of the lipid nanoparticles. For instance, membrane integrity studies using a hydrophilic marker (e.g. calcein) would benefit the manuscript.

Answer: Thank you for your positive appreciation of our paper and your questions.

Experiments with the hydrophilic calcein dye are valuable for evaluation of the membrane integrity of liposomes. Here, the cubosome (PL-C) and hexosome (PL-H) type of LNPs involve open channel networks and cannot encapsulate calcein in the aqueous compartments, because these compartments are not closed but open (scheme in Fig.1) and will not retain calcein inside the nanoparticles.

The PL-V carriers might be able to encapsulate calcein, but they are not suitable for time-resolved studies.

For the *in vitro* treatment study performed here, we always used freshly prepared LNPs (PL-C, PL-H and PL-V). In fact, PUFA-plasmalogen lipids can form oxidation products upon storage and PL encapsulation into LNPs helps to benefit from the PL therapeutic effects.

In our rationale, it was expected that the LNPs may protect the bioactive plasmalogens from oxidation or degradation in order to be administered to the PD model.

We plan a separate manuscript on the structural organization of PUFA-plasmalogen-based self-assembled nanocarriers, which will pay attention to the parameters, which are crucial for the LNPs stability.

In the revised Supplementary Information file, we have included data regarding the stability of the nanoparticle populations over time, as determined by quasi-elastic light scattering (Fig.5). All three formulations demonstrate size stability over a period of 10 days. While DLS is particularly well-suited for measuring the size of spherical particles, it can also provide size information for particles with non-spherical shapes. However, the accuracy of the measurements may be influenced by the shape and aspect ratio of the particles. In the case of non-spherical particles, DLS may provide an average size or an equivalent hydrodynamic radius that best represents the particle's behavior in a liquid medium. That's why other techniques like SAXS and cryo-TEM have been employed to obtain size and shape information.

Figure S5. Nanoparticle sizes determined for the three plasmalogen-based formulations at 25°C (n = 3).

In our opinion, the preservation of the LNP size and structure over time does not guarantee the preservation of the biological activity of PL in aged formulations. Therefore, further research will be needed on the relationship between the LNP stability over time and the biological efficacy of PL formulations stored for different times before use.

Microcalorimetry studies would offer important information on the miscibility of the actives with lipids and would further augment the results obtained from SAXS and Cryo-TEM studies. The SAXS studies were performed at 25°C and not at 37°C which is closer to the in vivo conditions. Moreover, time resolved studies would benefit the manuscript shedding light to the behavior of the formulation over time. Please comment on this.

Answer: Thank you for your suggestions.

Considering the properties of the investigated LNPs and nanoassemblies, we envision the intranasal delivery route for the next in-vivo test and their potential therapeutic applications.

The recommended temperature for the samples during intra-nasal delivery is usually around room temperature (20-25 °C) as in our SAXS investigation.

Intra-nasal delivery typically aims to maintain a relatively lower temperature rather than the body's temperature (37 °C). In fact, administration of substances at body temperature in the nasal cavity may cause discomfort or irritation⁷.

We do agree that the proposed experiments with additional techniques are in principal very helpful for the comprehension of drug delivery outcomes. However, considering that optimal effectiveness and safety of our formulations for intra-nasal delivery is expected without pre-

heating of the formulations, we did not plan microcalorimetry and SAXS studies over a temperature range for the purposes of this work.

Our comments in the revised version are:

“Moreover, nanomedicine may offer new strategies for delivering to the brain and enhancing the therapeutic effects.”

“An additional advantage of the investigated lipid-peptide nanosystem (constituted of mucoadhesive amphiphiles and cell-penetrating therapeutics) is that it can be delivered locally, for example by intranasal administration. This prospective platform should aim at nanomedicine-triggered neuronal survival pathways upon non-invasive drug delivery by LNPs.”

References

1. Thomson, A. C. *et al.* The Effects of Serum Removal on Gene Expression and Morphological Plasticity Markers in Differentiated SH-SY5Y Cells. *Cell Mol Neurobiol* **42**, 1829–1839 (2022).
2. Shipley, M. M., Mangold, C. A. & Szpara, M. L. Differentiation of the SH-SY5Y Human Neuroblastoma Cell Line. *J Vis Exp* **108**, 53193 (2016).
3. Guerzoni, L. P. B., Nicolas, V. & Angelova, A. In Vitro Modulation of TrkB Receptor Signaling upon Sequential Delivery of Curcumin-DHA Loaded Carriers Towards Promoting Neuronal Survival. *Pharm Res* **34**, 492–505 (2017).
4. Hossain, Md. S., Mineno, K. & Katafuchi, T. Neuronal Orphan G-Protein Coupled Receptor Proteins Mediate Plasmalogens-Induced Activation of ERK and Akt Signaling. *PLoS ONE* **11**, e0150846 (2016).
5. Encinas, M. *et al.* Sequential Treatment of SH-SY5Y Cells with Retinoic Acid and Brain-Derived Neurotrophic Factor Gives Rise to Fully Differentiated, Neurotrophic Factor-Dependent, Human Neuron-Like Cells. *Journal of Neurochemistry* **75**, 991–1003 (2000).
6. Yang, T.-X. *et al.* EPA-enriched plasmalogen attenuates the cytotoxic effects of LPS-stimulated microglia on the SH-SY5Y neuronal cell line. *Brain Research Bulletin* **186**, 143–152 (2022).
7. Elad, D., Wolf, M. & Keck, T. Air-conditioning in the human nasal cavity. *Respiratory Physiology & Neurobiology* **163**, 121–127 (2008).

REVIEWERS' COMMENTS:

Reviewer #1 (Remarks to the Author):

The authors have made the necessary changes accordingly.

Reviewer #2 (Remarks to the Author):

Most of my concerns have been addressed in the revised manuscript, however, I still have only one concern.

Minor point:

Please describe the number of the samples in Figures S2 - S4.

Reviewer #3 (Remarks to the Author):

The authors have sufficiently addressed my concerns. I recommend the acceptance of the manuscript

Reply to Reviewer's comments

Manuscript ID: COMMSCHEM-23-0328A

Title: *Sustained CREB Phosphorylation by Lipid-Peptide Liquid Crystalline Nanoassemblies*

We thank **Reviewer #1** and **Reviewer #3** for recommending the acceptance of the manuscript.

Reviewer #2

Most of my concerns have been addressed in the revised manuscript, however, I still have only one concern.

Minor point:

Please describe the number of the samples in Figures S2 - S4

Answer: Thank you for your comment. We added the requested information to the revised *Supporting Information* (Figures S2 - S4).

Figure S2. Effect of SH-SY5Y cellular differentiation and starvation on the fold changes in phosphorylated CREB (pCREB) levels before the oxidative damage by 6-OHDA. The cell culture conditions correspond to:

- non-differentiated SH-SY5Y cells grown in a medium containing FBS, *i.e.* no RA/FBS(+);
- non-differentiated SH-SY5Y cells exposed for 24h to FBS-free medium, *i.e.* no RA/FBS(-);
- RA-differentiated SH-SY5Y cells grown in FBS-containing complete medium for 5 days, *i.e.* RA/FBS(+);
- RA-differentiated SH-SY5Y cells exposed for 24h to FBS-free medium, *i.e.* RA/FBS(-).

Phosphorylated CREB was quantified by ELISA (n = 3). Using the Prism software, the Dunnett test was used for multiple group comparison, while the Student t-test was employed for comparing two groups. Statistical significance is indicated as * $P \leq 0.05$, ** $P \leq 0.01$, and *** $P \leq 0.001$.

Figure S3. Fold changes in the phosphorylation levels of CREB, AKT, and ERK proteins following 6-OHDA (30 min) treatment of 24h-starved differentiated SH-SY5Y cells, quantified by ELISA assays. The control RA/FBS(-) corresponds to RA-differentiated SH-SY5Y cells exposed for 24h to an FBS-free medium (n=3). Using the Prism software, the Dunnett test was used for multiple group comparison, while the Student t-test was employed for comparing two groups. Statistical significance is indicated as * $P \leq 0.05$, ** $P \leq 0.01$, and *** $P \leq 0.001$.

Figure S4. Fold changes in the phosphorylation levels of CREB protein (pCREB) after exposure to 6-OHDA (30 min, 200 μ M) or LNPs (24h, 10 μ M) incubation. Nanoformulations of PL-V, PL-H, and PL-C are compared. Protein phosphorylation levels were quantified using an ELSA assay with regard to an RA/FBS(-) control sample (i.e. differentiated SH-SY5Y cells deprived of serum for 24h) (n=3). Using the Prism software, the Dunnett test was used for multiple group comparison, while the Student t-test was employed for comparing two groups. Statistical significance is indicated as * $P \leq 0.05$, ** $P \leq 0.01$, and *** $P \leq 0.001$.